# A leafhopper saliva protein mediates horizontal transmission of viral pathogens from insect vectors into rice phloem

Wei Wu [1,2,3], Ge Yi[1,3], Xinwei Lv[1], Qianzhuo Mao[1,2] & Taiyun Wei [1✉]

Numerous insects transmit viruses together with saliva to plant phloem, but the roles of saliva components remain elusive. Here, we report that calcium-binding protein (CBP), a universal insect saliva protein, is modified to benefit horizontal transmission of a devastating rice reovirus into plant phloem. CBP effectively competes with virus-induced filaments to target and traverse actin-based apical plasmalemma into saliva-stored cavities in salivary glands of leafhopper vector. Thus, the inhibition of CBP expression by viral infection facilitates filament-mediated viral secretion into salivary cavities and then into plant phloem. Furthermore, virus-mediated reduction of CBP secretion causes an increase of cytosolic $Ca^{2+}$ levels in rice, triggering substantial callose deposition and $H_2O_2$ production. Thus, viruliferous vectors encounter stronger feeding barriers, probe more frequently, and secrete more saliva into plants, ultimately enhancing viral transmission. We thus conclude that the inhibition of CBP secretion facilitates viral secretion and increases host defense response to benefit viral transmission.

[1] Vector-borne Virus Research Center, State Key Laboratory of Ecological Pest Control for Fujian and Taiwan Crops, Fujian Agriculture and Forestry University, Fuzhou, Fujian 350002, China. [2] State Key Laboratory for Managing Biotic and Chemical Threats to the Quality and Safety of Agro-products, Key Laboratory of Biotechnology in Plant Protection of Ministry of Agriculture and Zhejiang Province, Institute of Plant Virology, Ningbo University, Ningbo, Zhejiang 315211, China. [3]These authors contributed equally: Wei Wu, Ge Yi. ✉email: weitaiyun@fafu.edu.cn

Many devastating plants, animals, and human viral pathogens are horizontally transmitted by arthropod insects together with saliva to susceptible hosts. Insect salivary gland cells are filled with abundant apical plasmalemma-lined cavities, where saliva is stored[1]. Arthropod-borne viruses (arboviruses) have to pass through the apical plasmalemma into insect salivary cavities, thereby moving with the salivary flow to establish the initial infection in hosts[2]. Insect saliva has important functions in modulating host–insect vector–arbovirus interactions. Mosquito saliva contains a repertoire of bioactive components that modulate the host's hemostasis and immune responses, thus facilitating blood-feeding and viral pathogen transmission[3,4]. Saliva proteins secreted by *Aedes aegypti*, such as CLIPA3 protease and a 34-kDa protein, can enhance Dengue virus (DENV) replication and suppress host innate immune responses[5,6]. Similarly, a saliva protein from *A. aegypti*, LTRIN, enhances Zika virus (ZIKV) infection and dissemination by interfering with nuclear factor-κB signaling and downstream inflammatory cytokine production[7]. A recent study by Sun et al. described *A. aegypti* venom allergen-1 as specifically expressed in the salivary glands, where it was found to activate autophagy in dendritic cells and monocytes, promoting DENV and ZIKV transmission[8]. Piercing-sucking insects such as leafhoppers, planthoppers, aphids, and whiteflies, can horizontally transmit viral pathogens together with their saliva to plant phloem[2]; however, how insect saliva functions in viral horizontal transmission into host phloem is still unclear.

Phloem functions as a long-distance transport system with highly evolved vascular tissue comprising sieve elements, companion cells, and parenchyma cells[9]. The stylets of piercing-sucking insects often traverse a long route through the intercellular spaces of plants to reach the phloem, and thus, the insects secrete watery saliva proteins to interfere with defense-associated callose deposition at sieve plates of plant phloem[10]. Insect vectors deliver viral pathogens into sieve cells via saliva[2]. Callose-associated sieve plate occlusion represents a potentially unique phloem defense strategy against viral pathogens and insects[11]. Sieve plate occlusion by callose sealing is likely dependent on $Ca^{2+}$ accumulation in sieve cells of plant phloem[11,12]. Phloem-feeding insects, such as brown planthopper *Nilaparvata lugens*, small brown planthopper *Laodelphax striatellus*, aphid *Megour aviciae*, whitefly *Bemisia tabaci*, and green rice leafhopper *Nephotettix cincticeps*, secrete calcium-binding proteins (CBPs) in watery saliva to decrease cytosolic $Ca^{2+}$ accumulation and prevent sieve cell occlusion in plant phloem, thus functioning as effectors to attenuate host plant defenses and improve insect feeding performance[13–17]. However, whether and how viruses mediate the regulation of CBP secretion from insect vectors to promote viral horizontal transmission into plant phloem remains unknown.

In this study, we examined the rice gall dwarf virus (RGDV), destructive plant reovirus, and its leafhopper vector, *Recilia dorsalis*, to determine how RGDV mediates the regulation of vector saliva components to benefit its own transmission into host phloem. RGDV was first described in 1979 in Thailand and caused severe epidemics in southern China and Southeast Asia[18,19]. During feeding, *R. dorsalis* ingests the phloem sap of RGDV-infected rice plants via its stylets and transmits virions into healthy plants through saliva secretion[1]. RGDV virions circulate in the insect vector body, enter the salivary glands, and then are released into the apical plasmalemma-lined cavities, where saliva is stored[1,20]. The cavity plasmalemma of vector salivary glands thus represents the last membrane barrier for viral transmission. RGDV first enters the cytoplasm of secretory cells in salivary glands and then initiates multiplication processes for the assembly of progeny virions[1]. The progeny virions of RGDV are associated with the filaments constructed by viral nonstructural protein Pns11 (Pns11 filaments) within the salivary glands[1]. Such virus-loaded Pns11 filaments attach to the actin-based apical plasmalemma and induce an exocytosis-like process for viral secretion into salivary cavities[1]. Viruliferous *R. dorsalis* encounters stronger physical barriers than nonviruliferous counterparts during feeding, ultimately prolonging salivary secretion and ingestion probing[21]. Such findings suggest that RGDV-mediated changes in insect feeding behavior potentially impede insects from continuously ingesting phloem sap and promote the secretion of more infectious virions from the salivary glands into rice phloem. However, whether RGDV infection modifies vector saliva components to benefit viral transmission remained unclear. In the present study, we revealed that RGDV infection could mediate the regulation of the saliva component CBP to benefit viral transmission into plant phloem, by promoting virions to traverse actin-based apical plasmalemma into saliva-stored cavities and increasing host plant defense responses.

## Results

### Sequences and expression characteristics of CBP in *R. dorsalis*.
Our preliminary transcriptome sequencing data analyses of *R. dorsalis* indicated that the expression of CBP in RGDV-infected salivary glands was significantly decreased relative to uninfected controls. To further investigate the expression characteristics of *R. dorsalis* CBP (RdCBP), the full-length RdCBP cDNA was first obtained (Supplementary Fig. 1, GenBank accession no. MZ476925). The 2150-bp RdCBP cDNA contains a 2061-bp open reading frame (ORF) that encodes a deduced 686-amino-acid (aa) protein (~79 kDa), which contains a single EF-hand domain, but no other domains of known function (Fig. 1a and Supplementary Fig. 1). The RT-qPCR assay revealed that RdCBP expression increased gradually with insect development until the fifth-instar nymph stage and then remained stable (Supplementary Fig. 2). A tissue-specific analysis indicated that RdCBP was specifically expressed in salivary glands but not in other organs, and thus the RdCBP antibody was specific (Fig. 1b, c). The salivary glands of leafhoppers consist of principal and accessory salivary glands, and the principal salivary glands (PSGs) are divided into six types of secretory cells (I–VI) (Fig. 1d). Immunofluorescence and immunoelectron microscopy revealed that RdCBP was specifically distributed in type III secretory cells of PSG, where it was localized in the secretory granules in the cytoplasm and accumulated abundantly in the salivary cavity (Fig. 1e–l).

### The inhibition of CBP expression by viral infection facilitates the secretion of RGDV from vector salivary glands.
We then examined the association between RdCBP expression and RGDV infection in salivary glands. RT-qPCR and western blot assays revealed that the expression of RdCBP in salivary glands was significantly decreased during RGDV infection (Fig. 2a, b). Immunofluorescence microscopy showed that the distribution areas of RdCBP were gradually reduced along with the viral infection process in the secretory cells of salivary glands (Fig. 2c–f and Supplementary Fig. 3a–c). After systemic infection with RGDV in the secretory cells, RdCBP was sparsely distributed throughout the cytoplasm but was still closely associated with the actin-based cavity plasmalemma or accumulated in the salivary cavities (Fig. 2g, h and Supplementary Fig. 3d–f). Immunoelectron microscopy confirmed that RdCBP was almost entirely absent in the cytoplasm of virus-infected secretory cells (Fig. 2i). Thus, viral infection strongly inhibited the expression of RdCBP in the salivary glands. Generally, viruses in the salivary cavities move with the salivary flow to the canal in the stylets and ultimately are injected into the sieve cells of plant phloem as viruliferous vectors feed on susceptible hosts. Western blot assay confirmed that the less levels of RdCBP were detected in rice seedlings that had been fed on by viruliferous leafhoppers for 2 days compared to those observed after being fed on by

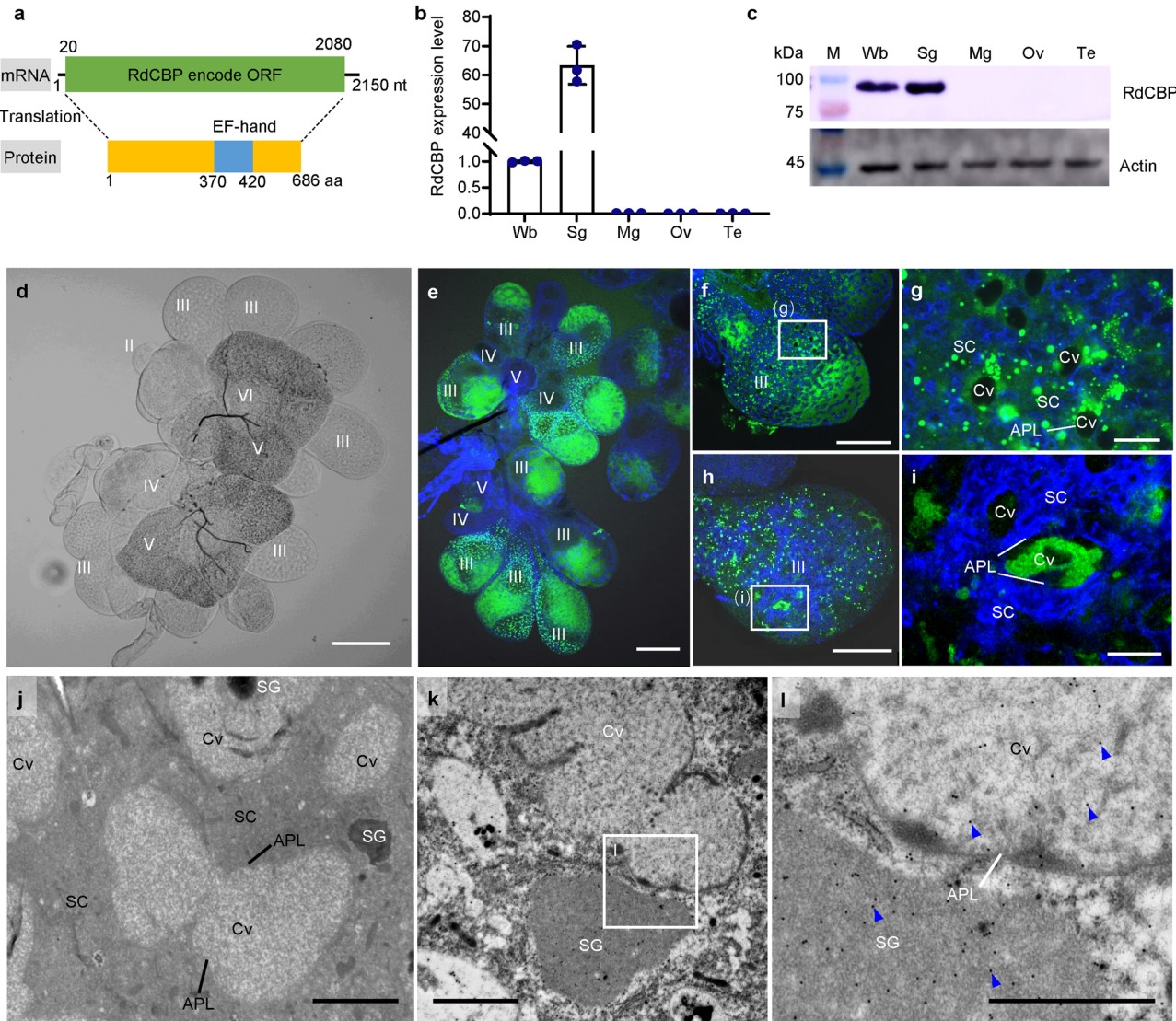

**Fig. 1 RdCBP is specifically expressed and stored in the saliva of type III secretory cells of *R. dorsalis* PSG. a** Diagram of the RdCBP gene. The RdCBP mRNA is 2150 bp in length and contains a 2061-bp ORF. The amino acid sequence of RdCBP is 686 aa long and contains a 51-aa EF-hand domain. **b, c** The mRNA and protein levels of RdCBP expression in different insect tissues were determined by RT-qPCR (**b**) and western blot (**c**) assay, respectively. Data in B were presented as mean ± SD values of three independent experiments. Significant differences were determined using Tukey's HSD test following an ANOVA. The proteins were detected by western blot assay by using RdCBP-specific or actin-specific IgGs. **d** The microscopic observation of the *R. dorsalis* PSGs, which have six types of secretory cells numbered from I-cells (I) to VI-cells (VI). **e–i** Immunofluorescence microscopy showing RdCBP was specifically expressed in the type III cells of PSGs (**e**). RdCBP was localized in the cytoplasm of type III secretory cells (**f, g**) and accumulated abundantly in the secretory cavities (**h, i**). Salivary glands were immunostained with RdCBP-FITC (green) and the actin dye phalloidin-Alexa Fluor 647 carboxylic acid (blue). **g, i** are the enlarged fields of the boxed areas in (**f, h**), respectively. **j** Salivary cavities in type III cells, as observed by electron microscopy. **k, l** Immunogold labeling of RdCBP in the cytoplasm and salivary cavities of type III secretory cells. PSGs were immunolabeled with RdCBP-specific IgG as a primary antibody, followed by treatment with a 15-nm gold particle-conjugated goat antibody against rabbit IgG as a secondary antibody. **l** is the enlarged field of the boxed area in (**k**). Blue arrows mark gold particles. Wb whole body, Sg salivary glands, Mg midgut, Ov ovary, Te testis, APL apical plasmalemma, Cv cavity, SC salivary cytoplasm, SG secretory granule. All images are representative of at least three replicates. Bars: **d, e** 100 μm; **f, h** 50 μm; **g, i** 10 μm; **j, k** 2 μm; **l** 1 μm.

nonviruliferous leafhoppers (Fig. 2j). Immunofluorescence microscopy revealed that RdCBP appeared in the sieve cells of rice phloem after being fed on by nonviruliferous leafhoppers for 2 days, confirming that RdCBP was secreted with the salivary flow into plant phloem via leafhopper feeding (Fig. 2k, l). Notably, RGDV was accompanied by RdCBP to secrete into the sieve cells of rice phloem as early as 2 days after being fed on by viruliferous leafhoppers (Fig. 2m).

We previously showed that RGDV Pns11-specific filaments in the cytoplasm of secretory cells were associated with actin-based

apical plasmalemma and then induced invagination to mediate viral dissemination into salivary cavities[1]. Immunofluorescence and immunoelectron microscopy further showed that, during viral infection of secretory cells, the distribution areas of RdCBP within the cytoplasm gradually decreased; however, the areas of apical plasmalemma-associated Pns11 filaments gradually increased (Fig. 2n–s, Supplementary Figs. 4, 5). All these results suggested that RGDV Pns11 competed with RdCBP to bind to actin-based apical plasmalemma. To test this hypothesis, we detected the competitive interactions among RdCBP, RGDV

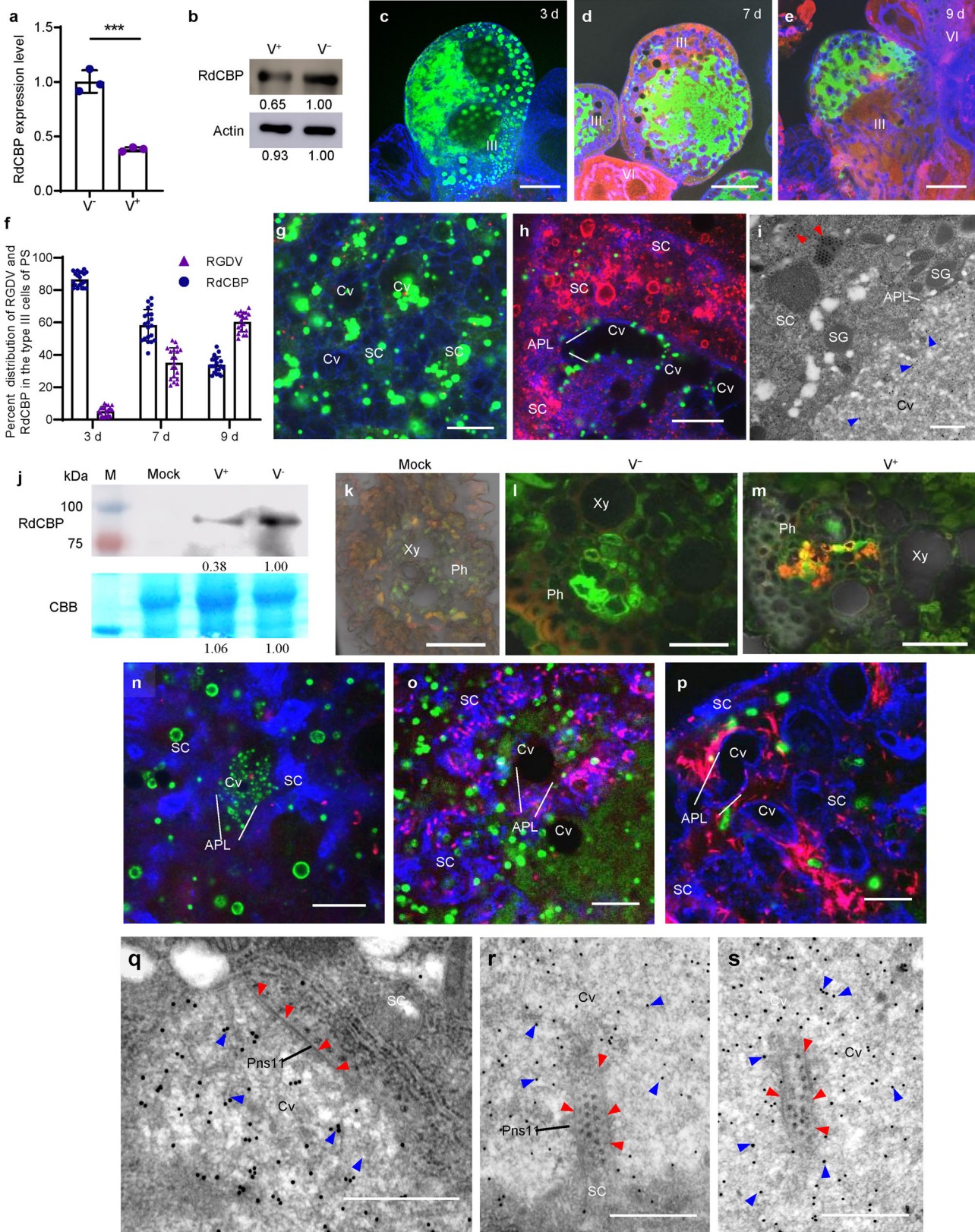

Pns11, and cytoplasmic actin of *R. dorsalis*, the main component of the apical plasmalemma. A yeast two-hybrid assay demonstrated that both Pns11 and RdCBP interacted with cytoplasmic actin (Fig. 3a). However, RdCBP was not observed to interact with Pns11 (Fig. 3a). A glutathione *S*-transferase (GST) pull-down assay confirmed that GST-fused actin was specifically bound to His-fused Pns11 and His-fused RdCBP, whereas control GST by itself did not (Fig. 3b). Furthermore, an in vitro competitive pull-down assay showed that the amounts of His-Pns11 pulled down by GST-actin increased as the amounts of His-RdCBP were reduced (Fig. 3c). In contrast, the increasing amounts of His-Pns11 did not affect the recovery of His-RdCBP

**Fig. 2 Reduced RdCBP secretion into rice phloem during RGDV secretion from vector salivary glands. a** RT-qPCR assay of RdCBP transcript levels in the salivary glands from nonviruliferous (V⁻) and viruliferous (V⁺) insects. Quantitative data from three independent experiments are presented as mean ± SD (error bars) values. Significant differences were tested using a two-tailed Student's *t*-test. ***$P < 0.0001$. **b** Western blot assay of the RdCBP accumulation levels in the salivary glands of nonviruliferous (V⁻) and viruliferous (V⁺) insects. RdCBP and actin were detected by using RdCBP- and actin-specific IgGs, respectively. Data were representative of three biological replicates. **c–h** Immunofluorescence assay showing that the distribution areas of RdCBP were gradually reduced in type III secretory cells of PSGs after 3 (**c**, **g**). 7 (**d**, **h**), and 9 (**e**) days post-microinjection of purified viruses, respectively. Virus-infected salivary glands were immunostained with RdCBP-FITC (green), virus-rhodamine (red), and phalloidin-Alexa Fluor 647 (blue), respectively. The average percentages of distribution areas of RdCBP and RGDV from 20 type III cells were counted using ImageJ (**f**). Bars represent mean ± SD values. **g**, **h** are the large images clearly showing the distribution of viruses and RdCBP in the cytoplasm or salivary cavities of secretory cells. **i** Immunoelectron microscopy showing the distribution of RdCBP in the cavities but not in the cytoplasm in virus-infected regions. Blue arrows mark gold particles. Red arrows mark viruses. **j** Western blot assay of RdCBP in rice plants that had been fed on by viruliferous (V⁺) or nonviruliferous (V⁻) leafhoppers. Rubisco large subunit was used as a loading control, as detected by staining with Coomassie Brilliant Blue (CBB). Mock, rice plants not subjected to leafhopper feeding. Data were representative of three biological replicates. M protein marker. **k–m** Immunofluorescence detection of RdCBP and RGDV in cross-sections prepared from leaf phloem infested with nonviruliferous or viruliferous *R. dorsalis*. Nonviruliferous (V⁻, **l**) or viruliferous (V⁺, **m**) *R. dorsalis* adults were fed on healthy rice plants for 3 days, after which plants were then processed for immunofluorescence with RdCBP-FITC (green) and virus-rhodamine (red). Non-infested healthy rice leaves served as a mock treatment (**k**). **n–p** Immunofluorescence assay showing that during viral infection, RGDV Pns11 filaments competed with RdCBP for release into salivary cavities. Virus-infected salivary glands dissected from insects after 3 (**n**), 7 (**o**), and 9 (**p**) days post-microinjection of purified viruses. Salivary glands were immunostained with Pns11-rhodamine (red), RdCBP-FITC (green), and phalloidin-Alexa Fluor 647 (blue). **q–s** Immunoelectron microscopy showing the distribution of RdCBP during RGDV Pns11 filament-mediated viral secretion into salivary cavities. Blue arrows mark gold particles. Red arrows mark Pns11 filaments. Virus-infected salivary glands in (**i**) and (**q–s**) were immunolabeled with RdCBP-specific IgG as a primary antibody, followed by treatment with 15-nm gold particle-conjugated goat antibody against rabbit IgG as a secondary antibody. The relative intensities of western blot bands in the analyses of different proteins for (**b**) and (**j**) are shown below. APL apical plasmalemma, Cv cavity, SC salivary cytoplasm, Xy xylem, Ph phloem. All images are representative of at least three replicates. Bars: **c–e** 50 μm; **k–m** 25 μm; **g**, **h**, **n–p** 10 μm; **i** 1 μm; **q–s** 500 nm.

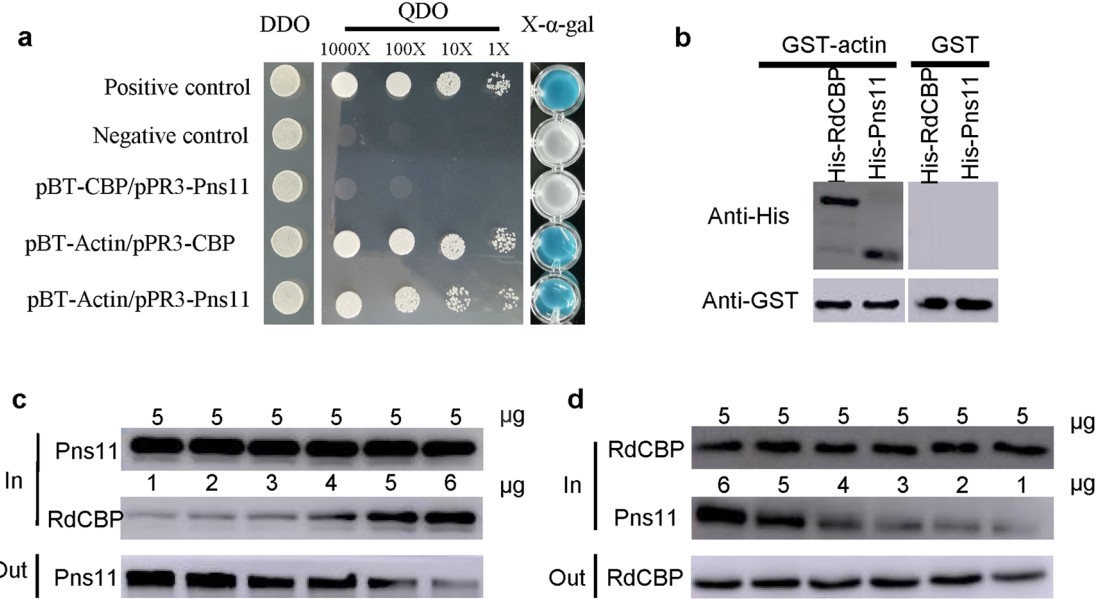

**Fig. 3 RdCBP–actin interaction is stronger than the Pns11–actin interaction. a** Interactions among RGDV Pns11, RdCBP, and cytoplasmic actin from *R. dorsalis* were detected by yeast two-hybrid assay. DDO, SD-Trp-Leu medium; QDO, SD-Trp-Leu-His-Ade medium. **b** GST pull-down assays were used to test the interactions of cytoplasmic actin with RdCBP and RGDV Pns11. Cytoplasmic actin was fused with GST as the bait protein. CBP and RGDV Pns11 were fused with His as the prey proteins. GST was used as a control. **c**, **d** The competitive Pull-down assays. **c** A fixed amount of His-Pns11 was mixed with different amounts of His-RdCBP and pulled down by GST-actin. **d** A fixed amount of His-RdCBP was mixed with different amounts of His-Pns11 and pulled down by GST-actin. Immunoblots were performed using anti-His or anti-GST antibodies to detect the associated proteins.

(Fig. 3d). These results indicated that RGDV Pns11 competed with RdCBP for direct binding to *R. dorsalis* cytoplasmic actin and that the RdCBP–actin interaction was stronger than the Pns11–actin interaction.

**The inhibition of CBP expression facilitates horizontal transmission of RGDV from viruliferous *R. dorsalis* into rice plants.** To further determine the role of RdCBP in the horizontal transmission of RGDV to rice plants, we used RNA interference

(RNAi) to knock down the expression of RdCBP. The third instar nymphs of *R. dorsalis* were microinjected with a mixture of the purified RGDV virions and dsRNAs targeting RdCBP or GFP genes (dsRdCBP or dsGFP). Immunofluorescence microscopy showed that RGDV established its initial infection of the dsGFP- and dsRdCBP-treated salivary glands at 5 and 3 days post-microinjection, respectively (Fig. 4a and Supplementary Fig. 6). RT-qPCR and western blot assays confirmed that the knockdown of RdCBP expression significantly increased the expression of the major outer capsid protein P8 of RGDV at the mRNA and

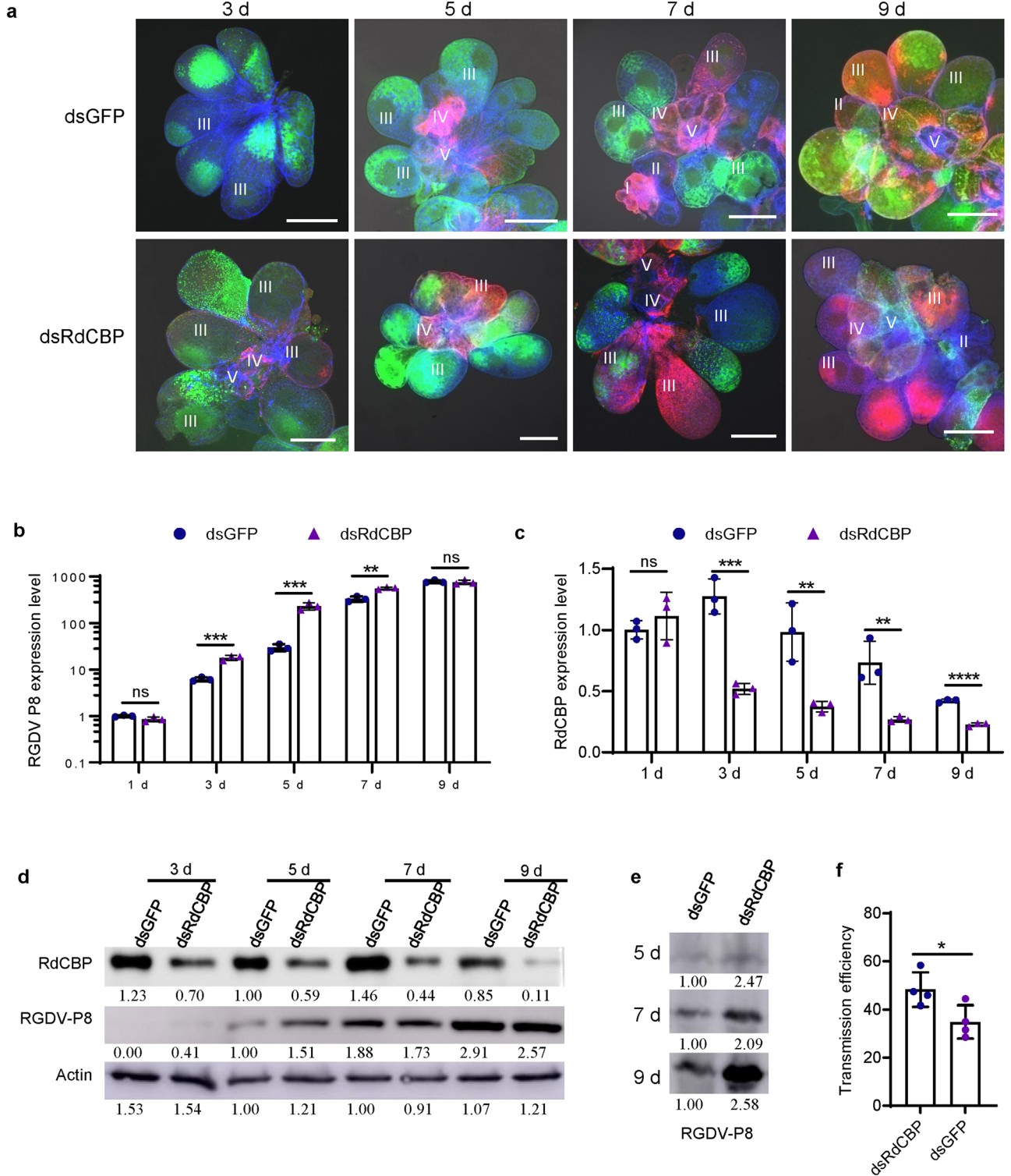

protein levels in the salivary glands prior to 9 days post-microinjection (Fig. 4b–d). Thus, the knockdown of RdCBP expression further promoted viral infection in salivary glands. In the secretory cells, RGDV effectively replicates and assembles progeny virions, which then are secreted into salivary cavities, facilitating viral transmission to plant hosts[1]. At 9 days post-microinjection, we observed that the knockdown of RdCBP expression effectively increased viral contents secreted from the salivary glands into rice phloem, ultimately promoting viral transmission by insect vectors into rice plants (Fig. 4e, f). Taken

together, these findings confirmed that the inhibition of RdCBP expression during RGDV infection in salivary glands facilitated viral secretion into salivary cavities and then into rice phloem.

**The increased cytosolic Ca²⁺ accumulation and callose deposition in rice plants after the inhibition of CBP expression in viruliferous *R. dorsalis*.** We then explored whether the reduction of RdCBP secretion into rice phloem during viruliferous *R. dorsalis* feeding influenced the cytosolic Ca²⁺ content by

**Fig. 4 The knockdown of RdCBP expression enhances viral infection and transmission from leafhoppers to rice plants. a** Immunofluorescence assay showing the distribution of RdCBP and RGDV in the salivary glands of *R. dorsalis* at different days post-microinjection of a mixture of purified viruses and dsRNAs. Salivary glands were immunostained with virus-rhodamine (red), RdCBP-FITC (green), and phalloidin-Alexa Fluor 647 (blue), and then processed for confocal microscopy. Bars, 100 μm. **b**, **c** Expression of RGDV P8 or RdCBP at the transcript levels in salivary glands at different days post-microinjection of a mixture of purified viruses and dsRNAs. EF1α was used as an internal control. Salivary glands were collected from 90 individual leafhoppers and randomly divided into three pools ($n = 30$ in each pool). Data from three independent experiments were presented as mean ± SD (error bars) values. Significant differences of feeding sites based on multiple Student's *t-test*. **d** The accumulation levels of RdCBP or RGDV P8 in salivary glands after different days post-microinjection of a mixture of purified viruses and dsRNAs were determined by western blot assay. Samples were detected with P8-, RdCBP-, and actin-specific antibodies. Actin served as the loading control. Salivary glands were collected from 90 individual leafhoppers and randomly divided into three pools ($n = 30$ in each pool). **e** The accumulation levels of viruses in rice stems after leafhopper feeding. Samples from rice stem fed by dsGFP- and dsRdCBP-treated leafhoppers for 72 h were detected with a P8-specific antibody. **f** Transmission efficiency of RGDV by *R. dorsalis* after dsRNA treatment. After 7 days post-microinjection, leafhoppers were transferred to three-leaf stage rice seedlings. Bars represent means ± SD. The relative intensities of bands in the analyses of different proteins for western blot assay in **d** and **e** are shown below. The significance of differences were determined using a two-tailed Student's *t*-test. *$P < 0.05$; **$P < 0.01$; ***$P < 0.001$; ****$P < 0.0001$; ns no significance.

confocal microscopy using the calcium probe Fluo-3/AM. Clear fluorescence was observed around leafhopper feeding sites, suggesting that leafhopper feeding led to the increase of cytosolic $Ca^{2+}$ contents in rice (Fig. 5a). Fluorescence intensities around the feeding sites of viruliferous leafhoppers were significantly higher than those of nonviruliferous leafhoppers at 1 and 3 h after infestation (Fig. 5a–c), suggesting that viruliferous leafhopper feeding caused the elevated $Ca^{2+}$ contents in the cells near feeding sites. Furthermore, we observed that dsRdCBP-treated nonviruliferous insect feeding also caused higher cytosolic $Ca^{2+}$ levels than those observed in dsGFP-treated controls (Fig. 5a–c). Additionally, the number of feeding sites of viruliferous and dsRdCBP-treated leafhoppers was significantly higher than those of nonviruliferous and dsGFP-treated leafhoppers at 1 and 3 h after infestation (Fig. 5b). Collectively, these results suggested that the reduction of RdCBP secretion into salivary cavities during viral infection effectively led to the increase of plant cytosolic $Ca^{2+}$ contents.

We then investigated whether the increase in plant cytosolic $Ca^{2+}$ accumulation during viruliferous insect feeding was related to sieve plate occlusion through callose deposition. The callose deposition on the sieve plates of rice plants was observed by staining with 0.1% aniline blue at 3 days after *R. dorsalis* feeding. Little or no callose deposition was found on the sieve plates in the leaf sheaths during feeding by nonviruliferous or dsGFP-treated nonviruliferous insects (Fig. 5d, e). The sieve plates of plants infested with viruliferous or dsRdCBP-treated nonviruliferous insects emitted stronger fluorescence than those of plants infested with dsGFP-treated nonviruliferous controls (Fig. 5d, e). Thus, our results confirmed that the reduction of RdCBP secretion into the salivary cavities during viruliferous insect feeding finally led to an increased callose deposition on the sieve plates at the stylets insertion points.

**The inhibition of CBP expression by viral infection affects *R. dorsalis* feeding behavior.** Phloem plugging, characterized by callose deposition on sieve plates in plants, is one important mechanism that prevents insects from ingesting phloem sap[10]. We used an electrical penetration graph (EPG) assay to investigate how the reduction of RdCBP secretion affected the feeding behavior of viruliferous *R. dorsalis* on rice plants. The EPG assay revealed that the frequency and mean duration of probes for non-ingestion (waveform A), active ingestion from the phloem with the stylets (waveform C), encountering obstacles during mechanical puncture with the stylets (waveform F), and continuous saliva secretion (waveform S) were significantly longer for the viruliferous group than for the nonviruliferous group (Fig. 6). Contrarily, the mean duration of probes for passive prolonged ingestion in the phloem (waveform E) was significantly shorter

for the viruliferous group than for the nonviruliferous group (Fig. 6). Furthermore, the EPG assay showed that dsRdCBP-treated nonviruliferous insect feeding also induced significantly longer waveform A, C, F, and S probes, but a shorter waveform E probe when compared with dsGFP-treated controls (Fig. 6). Together, all these results indicated that the inhibition of RdCBP expression potentially caused viruliferous vectors to require more probing attempts in the process of feeding and encounter more obstacles in the process of reaching the phloem. Thus, viruliferous vectors prolonged salivary secretion and active ingestion probing, further confirming the above result that viruliferous and dsRdCBP-treated leafhoppers established more number of feeding sites (Fig. 5b).

**The inhibition of CBP expression by viruliferous *R. dorsalis* promotes the production of $H_2O_2$ in rice plants.** In plants, calcium ions are a ubiquitous intracellular second messenger involved in plant defense response signaling pathways[13,14]. The 1-aminocyclopropane-1-carboxylic acid (ACC), jasmonic acid (JA), jasmonoyl-isoleucine (JA-Ile), salicylic acid (SA), and hydrogen peroxide ($H_2O_2$) signaling pathways have been reported to play central roles in plant defense responses[15,16]. Therefore, we examined whether the inhibition of RdCBP secretion during viruliferous *R. dorsalis* feeding influenced the biosynthesis of these phytohormones and signals in rice plants. Our results showed that the levels of ACC, JA, JA-ILE, and SA were similar between rice plants infested with viruliferous and nonviruliferous leafhoppers, whereas $H_2O_2$ levels were significantly higher in plants infested with viruliferous leafhoppers relative to non-viruliferous leafhoppers (Fig. 7a–e). Compared with dsGFP-treated control, dsRdCBP-treated nonviruliferous insect feeding did not alter the levels of these defense-related phytohormones and signals, except the case of $H_2O_2$ whose levels were enhanced at 3, 6, and 24 h after infestation (Fig. 7a–e). Together, our data demonstrated that the inhibition of RdCBP expression by vir-uliferous *R. dorsalis* facilitated the production of $H_2O_2$ in rice plants.

## Discussion

Arboviruses have co-evolved with insect vectors to optimize their horizontal transmission to hosts[7]. Insect saliva components are believed to modulate host immune responses and facilitate the horizontal transmission of arboviruses to hosts[5–8], however, the functional roles of insect saliva in viral horizontal transmission into plant phloem are still unclear. In this study, we identify and characterize a 79-kDa CBP with the EF-hand motif, which is specifically expressed and stored in the saliva of salivary gland type III cells of *R. dorsalis*. We further elucidate that the

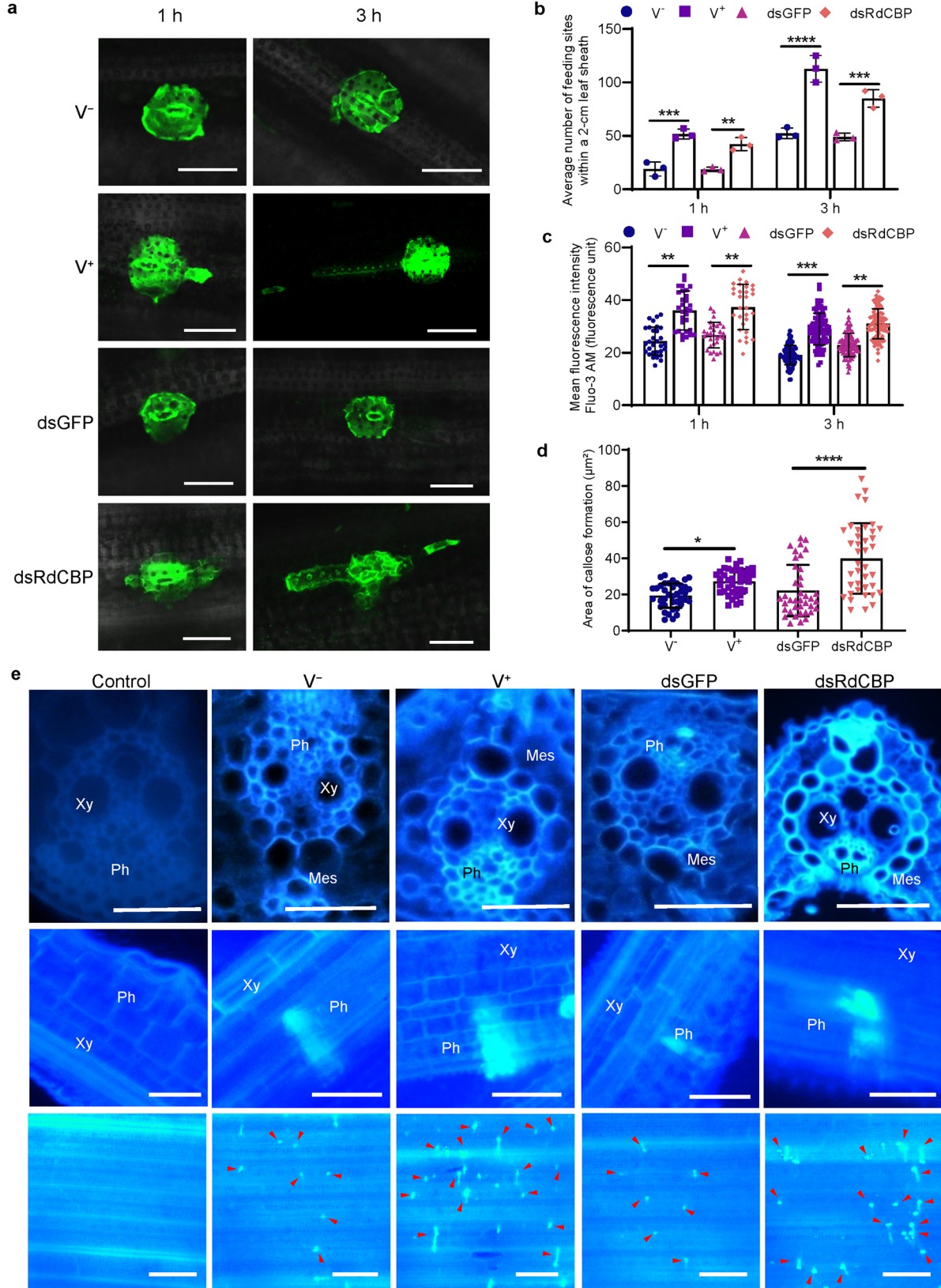

inhibition of CBP expression by RGDV infection promotes virions to traverse actin-based apical plasmalemma into saliva-stored cavities and increases host plant defense responses, which finally benefits viral horizontal transmission into plant phloem.

Type III secretory cells are the largest cells of *R. dorsalis* PSGs, and their cytoplasm is largely occupied by intracellular salivary

cavities in which saliva is stored[22]. The apical plasmalemma of the salivary cavities forms a loose network and provides large surface areas in the secretory cells. CBP directly attaches to the actin-based apical plasmalemma and then is secreted as a major salivary component during feeding[13–15]. RGDV infection induces the formation of virus-associated filaments constructed by viral

**Fig. 5 The reduction of RdCBP secretion into rice phloem during viruliferous *R. dorsalis* feeding causes elevated cytosolic Ca²⁺ levels and activates callose deposition. a** Fluochemical intracellular Ca²⁺ determination in leaves infested by nonviruliferous (V⁻) or viruliferous (V⁺) *R. dorsalis* adults, or infested by dsGFP- or dsRdCBP-treated nonviruliferous *R. dorsalis* adults. The green fluorescence refers to the binding of Fluo-3 AM with Ca²⁺. A portion of rice leaf infested by insects for 1 or 3 h was then incubated with 5 µM Fluo-3 AM solution. Red arrows mark feeding sites. Blue arrows mark Ca²⁺ signal transport along phloem. Bars, 25 µm. **b** Average number of feeding sites within a 2-cm leaf sheath infested by insects for 1 or 3 h. **c** Quantification of calcium levels as arbitrary units. Feeding sites of 30 randomly selected were observed by confocal microscopy, and the fluorescence intensity was calculated by Image J software. **d** The average areas of sieve plates with callose deposition in insect-infested leaf sheaths were counted using ImageJ. Error bars denote mean ± SD values of sieve plates with callose deposition observed in 40 cross-sections. **e** The callose deposition in the feeding site was visualized using bright blue fluorescence of cross-sections (top), longitudinal sections (middle), and whole leaf sections (bottom) prepared from leaf phloem infested with nonviruliferous (V⁻) or viruliferous (V⁺) *R. dorsalis* adults or infested by dsGFP- or dsRdCBP-treated nonviruliferous *R. dorsalis* adults. Samples without insect infestation served as the control. The thin sections were stained with 0.1% aniline blue at 3 days after *R. dorsalis* feeding and examined under a fluorescence microscope. Arrows indicate the bright blue fluorescence. Xy xylem, Ph phloem, Mes mesophyll. Bars represent means ± SD values. Significance was tested using one-way ANOVA. *P < 0.05; **P < 0.01; ***P < 0.001; ****P < 0.0001; ns no significance. Bars: **a** 25 µm; top and middle panels in **e** 50 µm; bottom panel in **e** 250 µm.

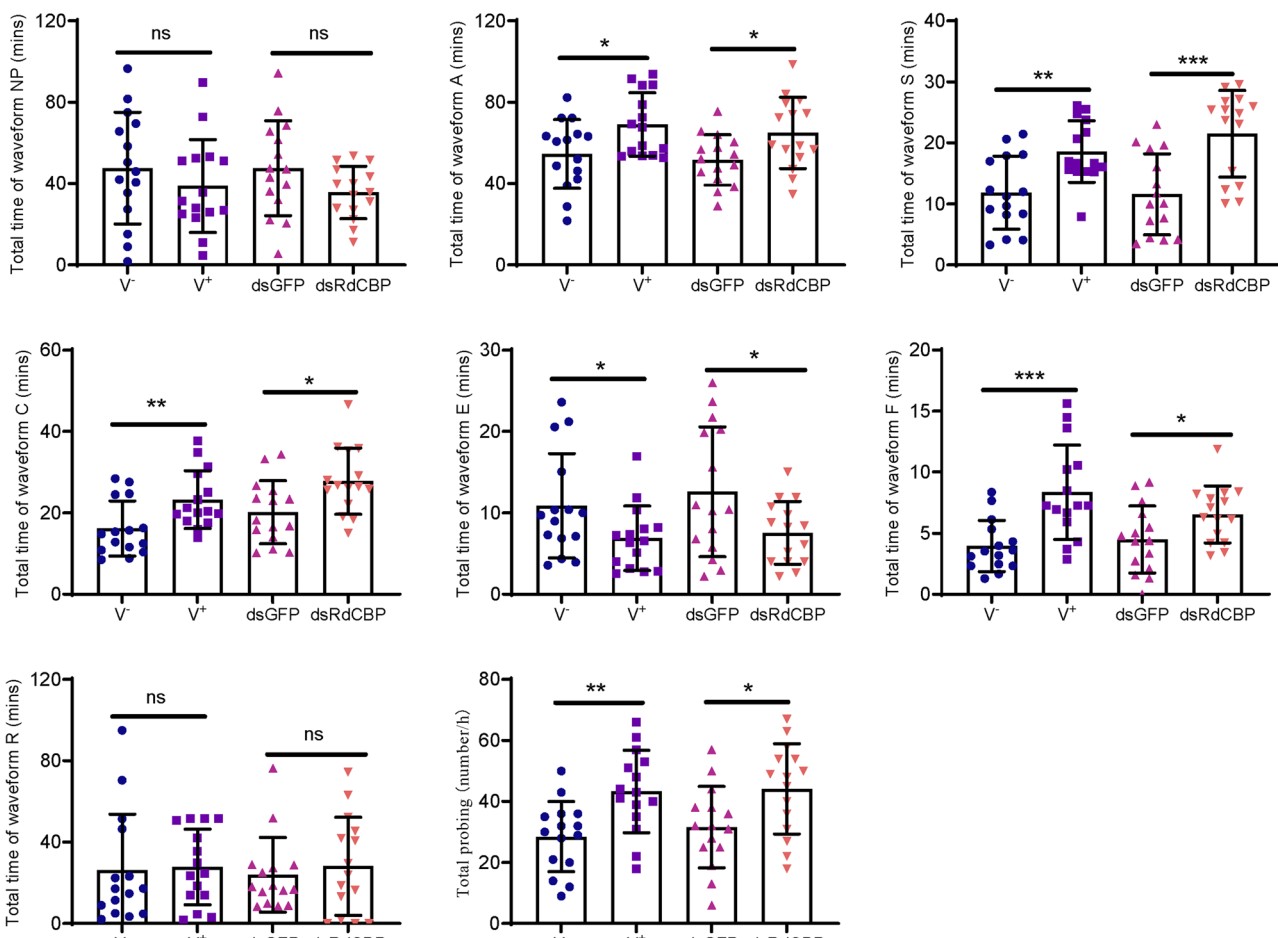

**Fig. 6 The reduction of RdCBP secretion into rice phloem during viruliferous *R. dorsalis* feeding is associated with an alteration in their feeding behavior, as detected by EPG assay.** Characterization of the EPG waveforms produced by *R. dorsalis* feeding on rice plants, including waveforms NP, A, S, C, E, F, R, and probing frequency. NP not probing. A stylets movement in the tissue. S intracellular salivation in mesophyll cells. C active ingestion from the phloem. E passive phloem sap ingestion. F obstacle waveform. R rest waveform. The data were electrically recorded during a 5-h feeding period for nonviruliferous or viruliferous *R. dorsalis* adults or dsGFP- or dsRdCBP-treated nonviruliferous *R. dorsalis* adults on healthy rice plants. Thirty leafhoppers in each treatment were prepared. Bars represent mean ± SD values. *P < 0.05; **P < 0.01; ***P < 0.001; ****P < 0.0001; ns no significance. Significance was tested using a one-way ANOVA.

nonstructural protein Pns11 (Pns11 filaments) within the salivary glands[1]. Such virus-loaded Pns11 filaments attach to actin-based apical plasmalemma and induce an exocytosis-like process for viral secretion into salivary cavities[1]. Both RGDV Pns11 and CBP compete to directly interact with the cytoplasmic actin, the main component of apical plasmalemma; however, the binding affinity of CBP with actin is stronger than that of Pns11 with actin. Thus,

the association of CBP with actin-based apical plasmalemma would competitively exclude the attachment of Pns11 filaments to apical plasmalemma. Viral infection strongly inhibits the expression and secretion of CBP from salivary glands. However, the knockdown of CBP expression using RNAi strategy further promotes viral accumulation and secretion from salivary glands. It is interesting that the knockdown of CBP expression in type III

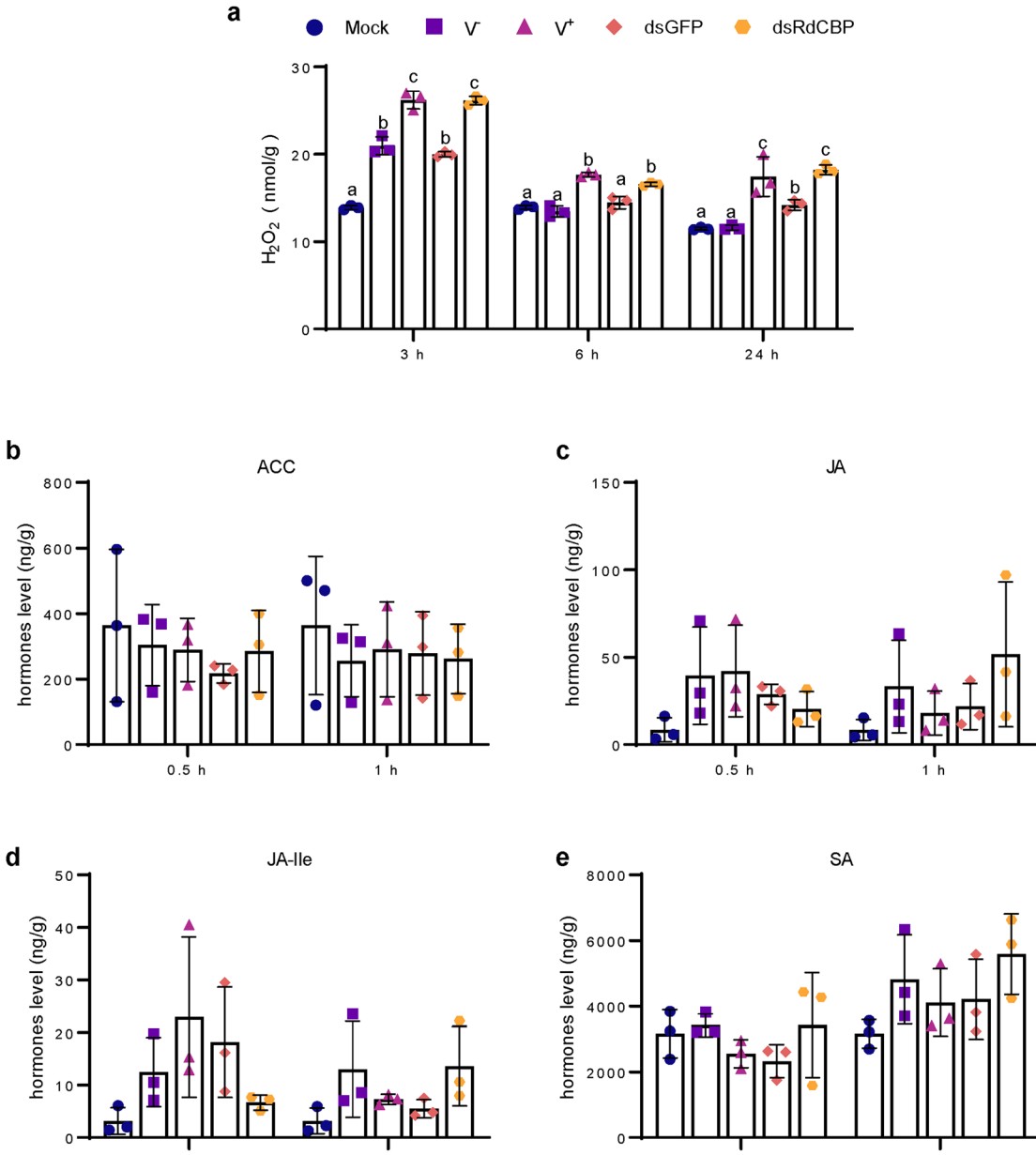

**Fig. 7 The reduction of RdCBP secretion into rice phloem during viruliferous *R. dorsalis* feeding affects plant defense responses.** The concentrations of $H_2O_2$ (**a**), 1-aminocyclopropane-1-carboxylic acid (ACC) (**b**), jasmonic acid (JA) (**c**), jasmonoyl-isoleucine (JA-Ile) (**d**), and salicylic acid (SA) (**e**) in leaves infested by viruliferous or nonviruliferous *R. dorsalis* adults or infested by dsGFP- or dsRdCBP-treated nonviruliferous *R. dorsalis* adults. The sample without insect infestation served as the control (mock). Data from three independent experiments were presented as mean ± SD values (error bars). Significance was tested using the one-way ANOVA.

secretory cells of PSG would enhance viral infection in the whole PSG, though the mechanism is still unknown. Thus, similar to small interfering RNA antiviral response[23], CBP may also play a crucial role in controlling the excessive viral replication in salivary. However, in virus-infected regions, CBP is almost restricted to the saliva but not present in the cytoplasm of secretory cells. As a propagative virus, RGDV must replicate to produce abundant progeny virions and Pns11 filaments in vector salivary glands[1]. Thus, without interference from CBP, abundant amounts of virus-loaded Pns11 filaments can effectively bind to apical plasmalemma to mediate viral secretion into the salivary cavities, guaranteeing efficient transmission of abundant virions from salivary glands into plant hosts (Fig. 8).

Plant phloem tissue represents a unique ecological niche for a variety of devastating pests that have evolved mechanisms to gain

access to it[10]. Phloem-feeding insects secrete watery and gelling saliva containing specific enzymes and/or effectors to interfere with plant defense responses in the phloem[10]. CBPs in the saliva function as effectors to attenuate host plant defenses in the phloem, thus improving insect feeding performance[13–17]. In this study, we further clarify that the inhibition of CBP secretion leads to increased host plant defense responses, which finally benefit viral transmission into rice phloem. Cytosolic $Ca^{2+}$ is an important mediator of sieve tube plugging in various plant species, including rice[17]. We find that the secretion of CBP from the salivary glands of viruliferous *R. dorsalis* into rice phloem is decreased, and thus the cytosolic $Ca^{2+}$ accumulation level and deposition of callose on the sieve plates are increased. Phloem plugging caused by callose deposition on sieve plates in plants is one of the most crucial defense mechanisms against the ingestion

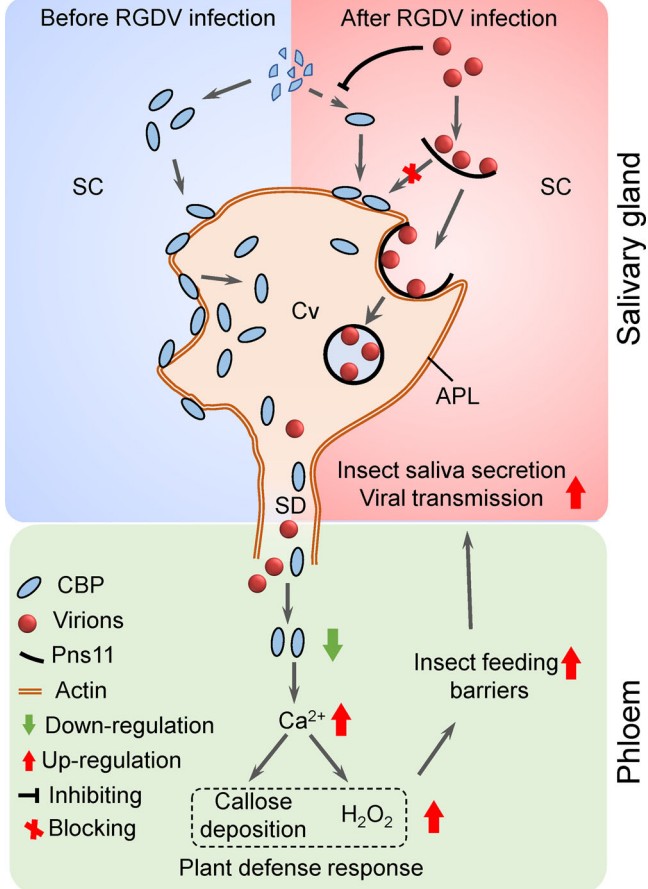

**Fig. 8 Proposed model of RGDV-mediated the inhibition of CBP expression to promote viral transmission by leafhopper vectors.** CBP can compete with Pns11 of RGDV to bind to cytoplasmic actin, the main component of the apical plasmalemma. Before viral infection of salivary glands, CBP directly attaches to the actin-based apical plasmalemma and then is secreted as a major salivary component. However, viral infection significantly inhibits CBP expression, and thus virus-loaded Pns11 filaments can attach to actin-based apical plasmalemma and induce an exocytosis-like process for viral secretion into salivary cavities. The reduction of CBP secretion increases cytosolic $Ca^{2+}$ levels in rice plants, triggering substantial callose deposition and $H_2O_2$ production. Thus, viruliferous vectors encounter stronger feeding barriers and would probe more frequently and secrete more saliva into rice plants, ultimately enhancing viral horizontal transmission. APL apical plasmalemma, SC secretory cytoplasm, Cv cavity, SD salivary duct.

of phloem sap by insects[12,13]. The EPG assay confirms that, relative to its nonviruliferous counterpart, viruliferous *R. dorsalis* encounters stronger physical barriers in the form of callose deposition, which prevents insects from continuously ingesting phloem sap. Cytosolic $Ca^{2+}$ is also a ubiquitous intracellular second messenger involved in plant defense response signaling pathways[24]. The increased cytosolic $Ca^{2+}$ accumulation level in rice plants during viruliferous *R. dorsalis* feeding facilitates the production of $H_2O_2$, but not other defense-related phytohormones and signals. It has been reported that the $H_2O_2$ signaling pathway positively regulates resistance in rice to phloem-feeding insects[25–27]. It is reasonable to deduce that the activated $H_2O_2$ signaling pathway in rice plants would also promote resistance to viruliferous *R. dorsalis*. Thus, the inhibition of CBP secretion ultimately leads viruliferous *R. dorsali* to encounter stronger barriers, including callose deposition and $H_2O_2$ production. This potentially stimulates viruliferous *R. dorsalis* to

probe more frequently and thus secrete more saliva into rice plants, thereby enhancing viral horizontal transmission. Together, we reveal a strategy by which CBP secretion into plant phloem is inhibited, thereby modulating host immune responses and facilitating pathogen transmission (Fig. 8).

Piercing-sucking insects, including leafhoppers, planthoppers, aphids, and whiteflies, horizontally transmit viral pathogens via saliva to plant phloem[28]. CBP is a universal component in the saliva of these phloem feeders, which would facilitate continuous feeding by inhibiting sieve tube plugging in plant phloem[13,25–27]. The green rice leafhoppers, aphids, and planthoppers secreted salivary CBPs with the conserved EF-hand motifs in their saliva, though the CBPs from different insect species have low amino acid sequence identities[13,29–31]. Our study reveals a mode of horizontal transmission of an arbovirus through inhibition of insect CBP secretion into plant phloem. We anticipate other arboviruses may have also evolved a similar strategy to promote their efficient viral horizontal transmission.

## Methods

**Insect, virus, and antibodies**. The leafhopper *R. dorsalis* adults were collected from rice fields in Xingning, Guangdong Province, China, and maintained in insect-proof greenhouses at $28 \pm 1$ °C under a 16:8 h light:dark photoperiod and $60 \pm 5\%$ relative humidity on TN-1 rice plants. The RGDV isolate was collected from infected rice fields in Xingning, Guangdong Province, China. The virus was transmitted to TN-1 rice plants by *R. dorsalis* following the methods described by Mao et al.[1].

Antibodies against intact virions, major outer capsid protein P8, and nonstructural protein Pns11 of RGDV were prepared as described previously[1]. The polypeptide of RdCBP (CMQQKTKSRSRRS) derived from *R. dorsalis* was conjugated to the carrier protein mariculture Keyhole Limpet Hemocyanin and injected into rabbits to generate the antibody against RdCBP. The antibody was produced by Genscript USA Innovation Company (Nanjing), which were approved by the Science Technology Department of Jiangsu Province, China with approval number SYXK (Su) 2018-0015. IgG was isolated from specific polyclonal antisera by using a protein A-Sepharose affinity column (Pierce). IgGs were conjugated directly to fluorescein isothiocyanate (FITC) or rhodamine according to the manufacturer's instructions.

**Expression analysis of RdCBP in the salivary glands of *R. dorsalis***. To analyze the expression levels of RdCBP in different tissues of *R. dorsalis*, the whole body, salivary gland, midgut, ovary, and testis were dissected from 200 newly emerged adult insects, total RNAs and proteins in different tissue were extracted using TRIzol Reagent (Invitrogen Cat#:15596026) and RIPA lysis buffer (Thermo Cat#:89900), respectively. The relative expression of RdCBP in different tissues of *R. dorsalis* adults was detected by RT-qPCR assay using a QuantStudio™ 5 Real-Time PCR system (Thermo Fisher Scientific). The detected transcript levels were normalized to the transcript level of the housekeeping gene elongation factor 1 alpha (EF1α) (GenBank accession number AB836665) and estimated by the $2^{-\triangle\triangle Ct}$ (cycle threshold) method[27]. To verify the expression patterns of RdCBP in different tissues of *R. dorsalis*, the total proteins were extracted from various insect tissues and then analyzed by western blot assay. We also detected the effects of RGDV infection on the protein levels of RdCBP expression in the salivary glands of viruliferous and nonviruliferous insects. In the corresponding western blot assay, RdCBP- and actin-specific IgG (1:1000 Sigma Cat#:SAB4502543) served as the primary antibodies, and goat anti-rabbit IgG-peroxidase served as the secondary antibody (1:5000 Sigma Cat#:A0545).

**Immunofluorescence staining of *R. dorsalis* salivary glands after viral infection**. Second instar nymphs of *R. dorsalis* were fed on diseased rice plants for 1 day and then transferred to healthy rice seedlings. At different days post-microinjection of purified virions, salivary glands from 50 *R. dorsalis* individuals were dissected, fixed in 4% paraformaldehyde in 0.1 M phosphate buffer (pH 7.2) for 2 h at room temperature. Salivary glands were washed with PBS and permeabilized in PBS that contained 0.1% Triton X-100 for 0.5 h at room temperature. After fixation, salivary glands were washed with PBS and immunolabeled with RdCBP-specific IgG conjugated to FITC (RdCBP-FITC) (1:50), virus- or Pns11-specific IgG conjugated to rhodamine (virus-rhodamine or Pns11-rhodamine) (1:50), and the actin dye (1:100 Invitrogen Cat#: A30107)[1]. Immunostained salivary glands were then imaged by a Leica TCS SP5 inverted confocal microscope[1]. The salivary glands dissected from *R. dorsalis* adults that fed on healthy rice plants were treated in exactly the same way and served as controls.

**Electron microscopy**. The salivary glands dissected from nonviruliferous or viruliferous *R. dorsalis* adults were fixed, dehydrated, and embedded, and the ultrathin sections were cut as previously described[1]. For immunoelectron microscopy, the

sections from salivary glands were immunolabeled with RdCBP- or Pns11-specific IgG as a primary antibody (1:50), followed by treatment with goat anti-rabbit/mouse IgG conjugated with gold particles as a secondary antibody (1:100 Abcam Cat#:G7652, G7402)[32]. Thin sections were examined with an H-7650 Hitachi transmission electron microscope (Hitachi, Tokyo, Japan).

**Yeast two-hybrid assay.** To test the interactions among RGDV Pns11, RdCBP, and cytoplasmic actin, a yeast two-hybrid assay was performed using a DUAL-membrane starter kit (Dualsystems Biotech, Zürich, Switzerland) (Supplementary Table 1). The cytoplasmic actin and RdCBP genes were cloned into the bait vector PBT-STE, while Pns11 and RdCBP genes were cloned into the prey vector pPR3-N, respectively. The bait and prey plasmids were co-transformed into *Saccharomyces cerevisiae* (strain NMY51). Plasmids pTSU2-APP and pNubG-Fe65 (positive control) or plasmid pPR3-N (negative control) were co-transformed into NMY51 yeast cells. All transformants were grown on synthetic dropout (SD) quadruple dropout (QDO) medium (SD/-Ade/-His/-Leu/-Trp) agar plates for 3 to 4 days at 30 °C. The strength of the protein–protein interaction between the bait and prey expressed in a particular clone was evaluated in a β-galactosidase assay using the HTX High-throughput β-Galactosidase Assay Kit (Dualsystems Biotech).

**GST pull-down and protein competitive interaction assay.** The cytoplasmic actin gene from *R. dorsalis* was cloned into PGEX- 3X for fusion with glutathione *S*-transferase (GST) (Supplementary Table 1). The RdCBP and Pns11 genes were cloned into pDEST17 for fusion with a His-tag. All recombinant proteins were expressed in *Escherichia coli* strain Rosetta and purified. GST-actin was bound to GST-Sepharose 4B beads (GE) for 3 h at 4 °C. Then, the mixture was centrifuged for 5 min, and the supernatant was discarded. His-tag fusion proteins were added to the beads and incubated for 2 h at 4 °C. After being centrifuged and washed five times with washing buffer (300 mM NaCl, 10 mM Na$_2$HPO$_3$, 2.7 mM KCl, and 1.7 M KH$_2$PO$_4$), the bead-bound proteins were separated by SDS-PAGE and detected by western blot with His-tag or GST-tag antibodies (1:1000 Sigma Cat#:SAB4301134, SAB4200692).

For the competitive pull-down assay, His-tag fusion proteins were purified using Ni-nitrilotriacetate (Ni-NTA) agarose (Qiagen). A fixed amount of His-Pns11 (5 μg) was mixed with different amounts of His-RdCBP (1, 2, 3, 4, 5, 6 μg) and pulled down by GST-actin. Alternatively, a fixed amount of His-RdCBP (5 μg) was mixed with different amounts of His-Pns11 (1, 2, 3, 4, 5, 6 μg) and pulled down by GST-actin. Beads were washed six times with washing buffer (300 mM NaCl, 10 mM Na$_2$HPO$_3$, 2.7 mM KCl, and 1.7 M KH$_2$PO$_4$). The associated proteins were separated by SDS-PAGE and detected by western blot with His-tag or GST-tag antibodies (1:1000 Sigma Cat#:SAB4301134, SAB4200692).

**Knocking down in vivo expression of RdCBP in *R. dorsalis*.** RNAi was performed to knock down the expression of RdCBP by microinjecting dsRdCBP into the abdomens of second instar nymphs. The dsRNAs targeting 500 bp regions of RdCBP or GFP were synthesized in vitro using the T7 RiboMAX Express RNAi System (Promega Cat#:P1700). The 50-s instar nymphs of *R. dorsalis* were microinjected with 30 nl of a mixture of purified viruses (0.01 μg μl$^{-1}$) and dsRdCBP or dsGFP (0.5 μg μl$^{-1}$) using a Nanoject II Auto-Nanoliter Injector (DRUMMOND). Thereafter, they were transferred to healthy rice seedlings for recovery. At 7 days post-microinjection, the individual insects were transferred into glass tubes that each contained a single rice seedling. The insects were allowed to feed on seedlings for 48 h and were individually analyzed by RT-PCR assay to confirm whether they were viruliferous or not. The seedlings inoculated with viruliferous leafhoppers were subjected to RT-PCR detection 20 days later. The test was conducted with more than three replicates. The transmission rate of RGDV by leafhoppers was calculated based on the RT-PCR results from the tested plants.

We then evaluated the effects of dsRdCBP treatment on efficient viral infection of salivary glands. The second instar nymphs of *R. dorsalis* were microinjected with a mixture of purified viruses and dsRdCBP or dsGFP. At 1, 3, 5, 7, and 9 days post-microinjection, the salivary glands from 30 leafhoppers were then fixed, immunolabelled with RdCBP-FITC, virus-rhodamine, and the actin dye phalloidin-Alexa Fluor 647 carboxylic acid (Invitrogen), and then processed for confocal microscopy. In addition, the levels of RdCBP and RGDV P8 were detected by RT-qPCR and western blot assays after treatment with dsRdCBP or dsGFP, respectively. Relative levels of gene expression were normalized to the expression level of EF1α and estimated by the 2$^{-\triangle\triangle Ct}$ (cycle threshold) method.

To assess the effects of dsRdCBP treatment on RGDV secretion during leafhopper feeding, the second instar nymphs were microinjected with a mixture of purified viruses and dsRNAs. At 5, 7, and 9 days post-microinjection, 30 leafhoppers were starved for 6 h and transferred to a cage to feed on a rice leaf for 24 h. These leaves were then collected, and their total proteins were extracted. The accumulated levels of RGDV P8 were detected by western blot assay.

**Western blot analysis and immunofluorescence staining of rice tissues.** Rice plant stems were confined individually within glass cylinders in which 50 dsGFP- and dsRdCBP-treated *R. dorsalis* were released, and 72 h later, the insects were removed. Total proteins were extracted using RIPA lysis buffer (Thermo Cat#:89900), and the accumulations of RdCBP and RGDV P8 were detected by western blot assay.

Hand-cut sections of tissues (about 30–50 mm in thickness) of the leaf sheaths of rice plants were collected at a specific time post-*R. dorsalis* infestation. These samples were then processed into 10–20-μm-thick sections using a Leica CM1900 cryostat. Tissue sections were fixed with 4% paraformaldehyde at room temperature for at least 12 h and permeabilized in 4% Triton X-100 for 6 h. Afterward, tissues were immunolabeled with virus-rhodamine and RdCBP-FITC, and then processed for confocal microscopy.

**EPG recordings.** To assess the effects of dsRdCBP treatment on the feeding behavior of leafhoppers, the second instar nymphs were microinjected with dsRdCBP or dsGFP, as described above. At 7 days post-microinjection, thirty leafhoppers were placed on a rice leaf in a cage, and their feeding behavior was recorded using a GiGA-8 DC electrical penetration graph amplifier system (Wageningen Agricultural University, Wageningen, The Netherlands)[33]. Meanwhile, the feeding behavior of viruliferous or nonviruliferous leafhoppers was also separately tested. The EPG recording was conducted in a quiet room with ambient conditions of 27 ± 2 °C and 70 ± 5% relative humidity. The recorded signals were analyzed using PROBE 3.4 software (Wageningen Agricultural University, Wageningen, The Netherlands).

**Evaluation of intracellular cytosolic Ca$^{2+}$ variation and callose deposition in rice plants.** To assess the effects of dsRdCBP treatment on the intracellular cytosolic Ca$^{2+}$ variation of rice plants after leafhopper feeding, 30 s instar nymphs were micro-injected with 15 ng of dsRdCBP or dsGFP. At 7 days post-microinjection, leafhoppers were allowed to feed on the seedlings in a leaf cage for 1 and 3 h, respectively. Meanwhile, ten viruliferous or nonviruliferous leafhoppers were each separately confined within leaf cages. The intracellular cytosolic Ca$^{2+}$ variation of rice plants was determined by using Fluo-3 AM (acetoxy-methyl ester of Fluo-3) (Invitrogen Cat#:F14218) as a Ca$^{2+}$-sensitive fluorescent indicator following a previously described method[34]. Subsequently, the leaves were examined by confocal microscopy. Micrographs were taken under identical conditions. The experiment was conducted in triplicate.

We then assessed the effects of dsRdCBP treatment on the callose deposition of rice plants after leafhopper feeding. Leaves of rice plants at the two-leaf stage were inoculated with ten insects (viruliferous, nonviruliferous, dsGFP-treated, or dsRdCBP-treated) for 6 h, cut into 0.3–0.5 cm pieces, and placed in Tissue-Tek OCT Compound (Sakura, Alphen aan den Rijn, The Netherlands). Samples were sliced into 10-μm sections using a Leica CM1900 frozen ultrathin slicer with a microtome and fixed on microscope slides, followed by overnight incubation in 100% ethanol solution. Thereafter, sections were stained with 0.1% aniline blue for 30 min and then examined by fluorescence microscopy. Micrographs were taken under identical conditions for all samples. Sieve plates with bright blue fluorescence were used to record callosic plates. For each sample, the area of callose in sieve plates was measured for 40 cross-sections.

**ACC, JA, JA-Ile, SA, and H$_2$O$_2$ analysis.** Rice leaves were harvested at different time points after insect feeding. ACC, JA, JA-Ile, and SA levels were analyzed by high-performance liquid chromatography–mass spectroscopy using labeled internal standards. The H$_2$O$_2$ concentrations in the rice sheaths were determined using an Amplex-Red Hydrogen Peroxide/Peroxidase Assay Kit (Thermo Cat#: A22188).

**Statistics and reproducibility.** All data were analyzed with SPSS (version 17.0; SPSS, USA). Percentage data were arcsine square root-transformed before analysis. Results were shown as mean value ± standard deviation (error bars) from at least three independent experiments. Multiple comparisons of the means were conducted using a one-way analysis of variance followed by Tukey's honestly significant difference test at the $P < 0.05$ significance level. Comparisons between two means were conducted using Student's *t*-test. The data were back-transformed after analysis in the text, figures, and tables.

**Reporting summary.** Further information on research design is available in the Nature Research Reporting Summary linked to this article.

## Data availability

The authors declare that all data supporting the findings of this study are available in the manuscript. Source data underlying the graphs are provided in Supplementary Data 1. Other relevant data are available from the corresponding author upon request.

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

## Acknowledgements

This work was supported by grants from the National Natural Science Foundation of China (Grants 31870149 and 31920103014).

## Author contributions

W.W. and T.W. conceived and designed the study. G.Y. and W.W. performed gene expression analysis and RNAi assay. W.W., G.Y., and Q.M. performed the protein interaction experiments. G.Y., W.W., and X.L. performed insect bioassays. W.W., G.Y., and T.W. analyzed the data and wrote the manuscript. T.W. organized the project. All authors read and approved the manuscript.

## Competing interests

The authors declare no competing interests.
