## [Peer Review File · Communications Biology]

Reviewers' comments:

Reviewer #1 (Remarks to the Author):

General comments

This work represents an incredible amount of valuable data on a very prominent question: that is the mechanisms of virus transmission by their arthropod vectors. It is indeed true that the specific role/impact of salivary proteins in this process has never been shown in plant viruses, where vector transmission is the major mode of transmission, and so where this question potentially concerns thousands of viral species. This paper shows that a saliva protein from an insect vector impacts on transmission by vectors, and that the virus can modify the expression of this protein in a way that has the potential to increase transmission. Manipulation of vectors by pathogens is a very important field of research and the amount of data reported is impressive, including all mechanisms explaining the observation.

The results are convincing, the experiments remarkably well performed and the approaches perfectly appropriate.

There are, however, a number of instances where the text should be modified, for clarity but not only. There is a tendency to over conclude, specifically when talking about processes that the virus would do to improve transmission. There is no will in viruses and I perfectly know that one may use such ease of language to simplify the message. However, there is no demonstration ever (to my knowledge) that a change of the vector (here decrease RdCBP secretion) induced by viral infection has actually been selected because it changes transmission. This is always an inference from the authors and it is very much the case here.

I have indicated below, in the specific comments, many instances where the text should be accordingly adjusted/improved.

Specific comments

Abstract:

-I do not understand why it is concluded that CBP mediates transmission of RGDV when it is the inhibition/reduction of its expression that has a positive effect on virus transmission.

-It is anticipated that it is a general viral strategy for vector born viruses... why? What justifies the "we thus anticipate" in the last sentence? As it is written the text does not support the idea that many other viruses would inhibit the expression of CBP. So, though this generalization is legitimate in the discussion part (the last paragraph of the discussion discusses this fine), I think it should not be stated like that in the abstract.

Introduction

-Line 56: How insect saliva functions in viral horizontal transmission is or "has been unclear" ?

-Line 60: there are rapid progresses in the understanding of phloem transport please complete or update REF 9

-Line 60: "stylet", here and elsewhere, should be plural (or use stylets bundle)

-Line 74: However, whether and how viruses mediate....

-Lines 78-79: the study of one viral/vector model cannot a priori be used to answer one question for all virus/vector couples. It can eventually be discussed later (again the last paragraph of the discussion is OK), but please change this sentence of the introduction to focus on this question for this model.

-Lines 92-94: this sentence is not relevant here, it does not provide relevant information and leads to auto-citation (Ref 21).

-Line 100: However, whether RGDV infections modifies....

-Lines 101-105: same comment as in Abstract. It is not CBP that promotes transmission through filaments formation and callose deposition but its absence or inhibition by the virus (if I understand well).

Results

-I do not understand Figure 2J, where is RGDV P8 ?

-Figure 2 Title is not correct. The fact that RdCBP decreases while virus accumulation increases does not demonstrate a link between the two. So it does not show that the former "facilitates" the latter.

-Figure 2 Legend line 748-749 remove repeats of Cv and SC

-Please indicate that Figure S3 and S4 are the same images as in Figure 2 but with split color channels

-Figure S4: Title is wrong. The fact that virus increases while RdCBP decreases shows the co-occurrence of the two phenomena but in no way does it show a competition.

-Lines 176-182: This text does not belong to the Result section but to the Discussion because it is speculation. It is already discussed in the Discussion section and so can simply be deleted here.

-Figure 4: I do not understand what means "Salivary glands were collected from 90 individuals and randomly divided into three pools (n=3 in each pool)". 3 pools of 3 individuals does not make 90?

-Issues related to Results presented in Figure 4 and S5:

i) RdCBP is expressed in type III cells of the salivary glands and it seems that the viral infection initiates in other cell types. So how can one explain that suppressing the expression of RdCBP via RNAi in type III cells facilitate the infection in other cell types? This could be discussed in the discussion section.

ii) Depletion of RdCBP greatly increases virus accumulation. This alone could explain that there is more virus in the saliva and better transmission. Thus, how much of the observed effect on transmission could be attributed to simple increase in virus accumulation versus to competition between viral filaments and RdCBP for actin binding. This must be discussed because it affects the conclusions on the mechanism through which the depletion of RdCBP actually favor horizontal transmission.

-Line 254: shorter E waveform for viruliferous insect is not significant, in Figure 261 ?

-Lines 258-264 belong to the discussion since it is a sort of interpretation/speculation from several experiments. By the way, that the inhibition of RdCBP expression by viral infection impacts on feeding behavior to facilitate transmission cannot be concluded. Indeed there is no demonstration that the inhibition of RdCBP expression is adaptive and has been selected specifically because it does change the behavior of the insect that should also be directly demonstrated to increase transmission. All this can only be speculated on the basis of the results and so it should be in the discussion section only.

Discussion

-lines 288-291: This sentence is explicitly evoking the role of vector saliva for plant pathogen transmission but all references cited (refs 5-8) do not address plant pathogens.

-Lines 293-295: virus mediate regulation of CBP secretion from insect vectors TO promote viral horizontal transmission... This is the very recurrent short cut that is not correct. Please remember that there is no demonstration that this phenomenon is adaptive/ Indeed, the virus reduces CBP secretion and this impacts on transmission but there is no proof that this is not simply fortuitous. This is true here and in very numerous papers from the corresponding literature. So please adapt the text so that what is concluded is directly supported by results, and so that speculation are

clearly exposed as such.

-Lines 308-318: Inhibition of CBP expression also increases viral accumulation, not solely NSP-11 filament secretion. So what part of the observed increase in virus transmission when silencing CBP gene is due to simple increase of viral accumulation or more specifically to reduced competition for actin filament and secretion into saliva should be discussed. So far this is omitted from the discussion.

-Lines 322-323: gelling saliva also contains proteins / effectors

-Lines 325-327: again please do not say that it has evolved TO inhibit CBP and thereby enhance transmission. There is only a demonstration that the virus inhibits CBP and that this consequently impacts on transmission. I repeat myself: That this observed phenomenon has specifically been selected because it increases transmission is not directly demonstrated.

-Lines 340-342: The fact that H₂O₂ impedes insect from continuously ingesting from the phloem is interesting. The discussion should cite the references and elaborate a bit more on the mechanisms or hypothesis described in the corresponding papers

-Last sentence: it is bizarre to imagine "targeting CBP secretion" to combat viral transmission after showing that CBP actually blocks transmission. Do you mean increasing the CBP secretion to combat transmission? If yes, please explicitly write it.

Materials and Methods

-Line 481: Thirty second instar nymphs? What does it mean? May be I am just ignorant on this point but over thirty larval stages seems a lot to me. Just checking.

-Line 492: same question

-Line 516 IDEM

Reviewer #2 (Remarks to the Author):

The authors report a dual action of virus infection on the saliva protein CBP. First, inhibition by an unidentified (viral?) mechanism lowers CBP levels in type-III salivary gland cells and promotes accumulation of virus via Pns11 filaments in salivary cavities. Second, lower CBP levels disturb normal 'stealthy' insect vector feeding behavior on rice plants and promote virus inoculation by provoking increased tissue punctures, caused by onset of plant defense reactions that interfere with feeding. The authors propose that by preventing calcium accumulation, CBP normally inhibits downstream plant defense responses against phloem feeding, in particular callose deposition and ROS formation, and that this inhibitory effect is elevated by the reduced CBP levels in infected insects. Indeed, lower CBP levels are found at feeding sites of viruliferous insects, and silencing CBP has the same effect.

This is a sound piece of work and the authors present ample evidence to back up their conclusions. The only 'brick' missing is how RGDV inhibits CBP, but the authors' intention is not to unveil this mechanism but to describe its effect on virus-vector-host interactions. The take home message 'viruses manipulate vector an insect saliva effector to induce insect feeding behavior conducive for virus transmission' is intriguing.

I have some concerns, though:

1. The authors claim that calcium accumulation is significantly higher at feeding sites from infected or CBP-silenced hoppers (lines 215ff). In my opinion, the images shown in Figure 5a alone do not allow to draw this conclusion. Quantification of the calcium signal taking into account fluorescence intensity/areas of feeding sites and using an adequate amount of feeding sites would be needed to back this claim. Further, as outlined below, the data do not allow to conclude whether you observe calcium signaling.

2. Whereas Figure 5c shows increased callose deposition in the vessels, I am not sure that the images show callose deposition on sieve plates, because they are practically invisible in transversal

sections. For this, one needs to observe longitudinal sections, which allow to localize and count callose deposition on sieve plates. This applies also to Figure 5d. Please clarify whether you observed and quantified 'general' callose deposition in vessels or sieve plate-specific callose deposition. I guess both deposition forms would interfere with feeding behavior.

3. The presentation of CBP is a bit short. It would be nice if you describe how you identified this protein and what is its relationship to other CBP from *R. d.* and other phloem feeders.

Other remarks:

Some confocal images are overexposed.

Line 60: It is not one stylet; the mouthparts of these insects are formed of 4 stylets that together form a needle-like structure comprising the food and saliva canal. So it is more adequate to refer to as 'stylets'.

Line 145: You mention here that in phloem infested with infected hoppers less CBP is detected but the western blot Figure 2j shows the contrary result. Please correct.

Line 203: You write here that more virus is detected in rice phloem when CBP-silenced infected hoppers are used to infest plants, but the legend to Figure 4e does not mention what the blot shows. Please correct.

Line 218: You see here Ca accumulation; whether these represent Ca signals can not be concluded from the data shown.

Line 248ff: The figure does not show frequencies (events/time) of waveforms, only their accumulated duration. However, Figure 5b shows that more feeding sites are established over time, which can be interpreted as a higher frequency of punctures. Please explain what exactly you mean by 'active ingestion from phloem'?

Line 320: Phloem sap is not really nutritionally rich; it is a rather unbalanced diet requiring adaptation of phloem feeders.

Lines 386-393: Please indicate briefly how you prepared total RNA and protein extracts from insects and/or plants, and the oligonucleotides used.

Line 401: Please indicate more in detail how you carried out immunofluorescence and acquired microscopic images or cite corresponding publications.

Line 421: Reference (1) is not very useful to describe the methodology because it refers to yet other references.

Line 451: Here you use a his and GST antibodies for protein detection, in the legends to the corresponding experiments you use GST, his, actin, CBP and Pns11 antibodies. Please verify.

Line 469: Are these final concentrations?

Line 790: (e) the label of the blots is missing.

Line 545: What do you mean by 'freezing glue'?

Line 559: Who is the provider of the kit and is it fluorescence-based or colorimetric?

Line 885: d-f. In 3Sd, actin and virus images are switched.

In some figures, 'levels' is misspelled, in Figure 7A 'mock' is misspelled.

Figure 1C: Please label with which antibody the blots were incubated.

Figure 2J: The text in the legend and in the blot diverge (actin in the blot, RGDV P8 in the legend). Why is the actin blot in blue? Please rectify.

Reviewer #3 (Remarks to the Author):

Insect vectors can transmit viral pathogens together with their saliva to plant phloem, but the roles of saliva components remain elusive. In this manuscript, Wu et al investigate the function of calcium-binding protein (CBP), a saliva protein of leafhopper, in the viral transmission by insect vectors. They found that the CBP competed with viral Pns11 filaments of rice gall dwarf virus (RGDV) to traverse the apical plasmalemma into saliva-stored cavities in salivary glands of leafhopper vectors. They further found that virus-caused reduction of CBP triggered substantial callose deposition and H₂O₂ production, which caused the stronger feeding barriers for viruliferous leafhopper vectors. More frequent probe and salivary secretion ultimately enhanced viral transmission.

I think this story is very interesting due to the relationship among salivary protein, pathogens, and insect behavior. These findings will benefit for understanding how salivary proteins can regulate the interaction between plant host, virus, and insect vector. of vector-borne viruses. However, I also have some questions that authors have to response.

1. Fig. 2q-s. Salivary glands were only immunolabeled with RdCBP-specific antibodies. However, authors did not show that the apical plasmalemma-associated filamentous structure was formed by RGDV Pns11. The immunoelectron microscopy of Pns11 is suggested.
2. Fig. 4, How did authors determine the injection doses of dsRNA in the RNAi experiment? Did different dose of dsRNA have different effect on RGDV infection in salivary glands?
3. Fig. 5, How many insects were recorded in the EPG experiment? Please show in the figure legend and Methods.
4. How RGDV replicated and assembled in the cells salivary glands? Authors should discuss this.

Minor comments:

1. The specificity of the RdCBP antibody should be shown in Materials and Methods.

Point by point to Comments

We would like to thank the editor and three expert reviewers for their insightful comments on the paper, as these comments led us to an improvement of the manuscript. Our revision reflects all editor's and reviewers' suggestions and comments. Detailed responses are given below.

Reviewers' comments:

Reviewer #1 (Remarks to the Author):

General comments

This work represents an incredible amount of valuable data on a very prominent question: that is the mechanisms of virus transmission by their arthropod vectors. It is indeed true that the specific role/impact of salivary proteins in this process has never been shown in plant viruses, where vector transmission is the major mode of transmission, and so where this question potentially concerns thousands of viral species. This paper shows that a saliva protein from an insect vector impacts on transmission by vectors, and that the virus can modify the expression of this protein in a way that has the potential to increase transmission. Manipulation of vectors by pathogens is a very important field of research and the amount of data reported is impressive, including all mechanisms explaining the observation.

The results are convincing, the experiments remarkably well performed and the approaches perfectly appropriate.

There are, however, a number of instances where the text should be modified, for clarity but not only. There is a tendency to over conclude, specifically when talking about processes that the virus would do to improve transmission. There is no will in viruses and I perfectly know that one may use such ease of language to simplify the message. However, there is no demonstration ever (to my knowledge) that a change of the vector (here decrease RdCBP secretion) induced by viral infection has actually been selected because it changes transmission. This is always an inference from the authors and it is very much the case here. I have indicated below, in the specific

comments, many instances where the text should be accordingly adjusted/improved.

Response: Thank you for your positive evaluation of our work.

Specific comments

Abstract:

-I do not understand why it is concluded that CBP mediates transmission of RGDV when it is the inhibition/reduction of its expression that has a positive effect on virus transmission.

Response: Thanks for this valuable comment. In the full text, including Title, Abstract, Results, Discussion and Figure legends, we have revised the relative sentences, and elucidate that the inhibition of CBP expression by RGDV infection promotes virions to traverse actin-based apical plasmalemma into saliva-stored cavities and increases host plant defense responses, which finally benefits viral horizontal transmission into plant phloem.

-It is anticipated that it is a general viral strategy for vector born viruses... why? What justifies the “we thus anticipate” in the last sentence? As it is written the text does not support the idea that many other viruses would inhibit the expression of CBP. So, though this generalization is legitimate in the discussion part (the last paragraph of the discussion discusses this fine), I think it should not be stated like that in the abstract.

Response: We have revised this sentence as “We thus conclude that the inhibition of CBP secretion facilitates viral secretion and increases host defense response to benefit viral transmission.” on lines 31-33.

Introduction

-Line 56: How insect saliva functions in viral horizontal transmission is or “has been unclear” ?

Response: We have revised this on lines 57-58.

-Line 60: there are rapid progresses in the understanding of phloem transport please

complete or update REF 9

Response: We have updated the Ref 9.

-Line 60: “stylet”, here and elsewhere, should be plural (or use stylets bundle)

Response: We have changed “stylet” to “stylets” in this manuscript.

-Line 74: However, whether and how viruses mediate....

Response: We have revised this on line 75.

-Lines 78-79: the study of one viral/vector model cannot a priori be used to answer one question for all virus/vector couples. It can eventually be discussed later (again the last paragraph of the discussion is OK), but please change this sentence of the introduction to focus on this question for this model.

Response: We have revised this sentence on lines 78-81.

-Lines 92-94: this sentence is not relevant here, it does not provide relevant information and leads to auto-citation (Ref 21).

Response: We have deleted this sentence.

-Line 100: However, whether RGDV infections modifies....

Response: We have revised this on line 100.

-Lines 101-105: same comment as in Abstract. It is not CBP that promotes transmission through filaments formation and callose deposition but its absence or inhibition by the virus (if I understand well).

Response: We have revised this sentence on lines 101-105.

Results

-I do not understand Figure 2J, where is RGDV P8 ?

Response: RGDV P8 is not detected in Figure 2J, which has been corrected in the text.

-Figure 2 Title is not correct. The fact that RdCBP decreases while virus accumulation increases does not demonstrate a link between the two. So it does not show that the former “facilitates” the latter.

Response: We have revised this on lines 725-726.

-Figure 2 Legend line 748-749 remove repeats of Cv and SC

Response: We have revised this in the Figure 2 legend.

-Please indicate that Figure S3 and S4 are the same images as in Figure 2 but with split color channels

Response: We have revised this in the Figures S3 and S4 legends.

-Figure S4: Title is wrong. The fact that virus increases while RdCBP decreases shows the co-occurrence of the two phenomena but in no way does it show a competition.

Response: We have revised this in the Figure S4 legend.

-Lines 176-182: This text does not belong to the Result section but to the Discussion because it is speculation. It is already discussed in the Discussion section and so can simply be deleted here.

Response: We have deleted these sentences in the revised version.

-Figure 4: I do not understand what means “Salivary glands were collected from 90

individuals and randomly divided into three pools (n=3 in each pool)". 3 pools of 3 individuals does not make 90?

Response: This is a description error, and it should be n=30, which has been corrected in the text

-Issues related to Results presented in Figure 4 and S5:

i) RdCBP is expressed in type III cells of the salivary glands and it seems that the viral infection initiates in other cell types. So how can one explain that suppressing the expression of RdCBP via RNAi in type III cells facilitate the infection in other cell types? This could be discussed in the discussion section.

ii) Depletion of RdCBP greatly increases virus accumulation. This alone could explain that there is more virus in the saliva and better transmission. Thus, how much of the observed effect on transmission could be attributed to simple increase in virus accumulation versus to competition between viral filaments and RdCBP for actin binding. This must be discussed because it affects the conclusions on the mechanism through which the depletion of RdCBP actually favor horizontal transmission.

Response: Viral infection strongly inhibits the expression and secretion of CBP from salivary glands. However, the knockdown of CBP expression using RNAi strategy further promotes viral accumulation and secretion from salivary glands. It is interesting that the knockdown of CBP expression in type III secretory cells of PSG would enhance viral infection in the whole PSG, though the mechanism is still unknown. Thus, similar to small interfering RNA antiviral response, CBP may also play a crucial role in controlling the excessive viral replication in salivary. However, in virus-infected regions, CBP is almost restricted to the saliva but not present in the cytoplasm of secretory cells. As a propagative virus, RGDV must replicate to produce abundant progeny virions and Pns11 filaments in vector salivary glands. Thus, without interference from CBP, abundant amounts of virus-loaded Pns11 filaments can effectively bind to apical plasmalemma to mediate viral secretion into the salivary cavities, guaranteeing efficient transmission of abundant virions from salivary glands into plant hosts.

We have discussed these on lines 309-323.

-Line 254: shorter E waveform for viruliferous insect is not significant, in Figure 261 ?

Response: Waveform E for viruliferous insect is significantly shorter than nonviruliferous control. We have corrected this in Figure 6.

-Lines 258-264 belong to the discussion since it is a sort of interpretation/speculation from several experiments. By the way, that the inhibition of RdCBP expression by viral infection impacts on feeding behavior to facilitate transmission cannot be concluded. Indeed there is no demonstration that the inhibition of RdCBP expression is adaptive and has been selected specifically because it does change the behavior of the insect that should also be directly demonstrated to increase transmission. All this can only be speculated on the basis of the results and so it should be in the discussion section only.

Response: We have deleted these sentences.

Discussion

-lines 288-291: This sentence is explicitly evoking the role of vector saliva for plant pathogen transmission but all references cited (refs 5-8) do not address plant pathogens.

Response: We have revised this sentence on lines 384-287.

-Lines 293-295: virus mediate regulation of CBP secretion from insect vectors TO promote viral horizontal transmission... This is the very recurrent short cut that is not correct. Please remember that there is no demonstration that this phenomenon is adaptive/ Indeed, the virus reduces CBP secretion and this impacts on transmission but there is no proof that this is not simply fortuitous. This is true here and in very numerous papers from the corresponding literature. So please adapt the text so that what is concluded is directly supported by results, and so that speculation are clearly exposed as such.

Response: We have revised this sentence as “We further elucidate that the inhibition of CBP expression by RGDV infection promotes virions to traverse actin-based apical

plasmalemma into saliva-stored cavities and increases host plant defense responses, which finally benefits viral horizontal transmission into plant phloem.” on lines 290-293.

-Lines 308-318: Inhibition of CBP expression also increases viral accumulation, not solely NSP-11 filament secretion. So what part of the observed increase in virus transmission when silencing CBP gene is due to simple increase of viral accumulation or more specifically to reduced competition for actin filament and secretion into saliva should be discussed. So far this is omitted from the discussion.

Response: Viral infection strongly inhibits the expression and secretion of CBP from salivary glands. However, the knockdown of CBP expression using RNAi strategy further promotes viral accumulation and secretion from salivary glands. It is interesting that the knockdown of CBP expression in type III secretory cells of PSG would enhance viral infection in the whole PSG, though the mechanism is still unknown. Thus, similar to small interfering RNA antiviral response, CBP may also play a crucial role in controlling the excessive viral replication in salivary. However, in virus-infected regions, CBP is almost restricted to the saliva but not present in the cytoplasm of secretory cells. As a propagative virus, RGDV must replicate to produce abundant progeny virions and Pns11 filaments in vector salivary glands. Thus, without interference from CBP, abundant amounts of virus-loaded Pns11 filaments can effectively bind to apical plasmalemma to mediate viral secretion into the salivary cavities, guaranteeing efficient transmission of abundant virions from salivary glands into plant hosts.

We have discussed these on lines 309-323.

-Lines 322-323: gelling saliva also contains proteins / effectors

Response: We have revised this on line 326-328.

-Lines 325-327: again please do not say that it has evolved TO inhibit CBP and thereby enhance transmission. There is only a demonstration that the virus inhibits CBP and that this consequently impacts on transmission. I repeat myself: That this observed phenomenon has specifically been selected because it increases transmission is not

directly demonstrated.

Response: We have revised this sentence as “In this study, we further clarify that the inhibition of CBP secretion leads to the increased host plant defense responses, which finally benefits viral transmission into rice phloem.” on lines 330-332.

-Lines 340-342: The fact that H₂O₂ impedes insect from continuously ingesting from the phloem is interesting. The discussion should cite the references and elaborate a bit more on the mechanisms or hypothesis described in the corresponding papers

Response: The increased cytosolic Ca²⁺ accumulation level in rice plants during viruliferous *R. dorsalis* feeding facilitates the production of H₂O₂, but not other defense-related phytohormones and signals. It has been reported that H₂O₂ signaling pathway positively regulates resistance in rice to phloem-feeding insects (Huang et al., 2019; Ji et al., 2021; Ye et al., 2017). It is reasonable to deduce that the activated H₂O₂ signaling pathway in rice plants would also promote resistance to viruliferous *R. dorsalis*. Thus, the inhibition of CBP secretion ultimately leads viruliferous *R. dorsali* to encounter stronger barriers, including callose deposition and H₂O₂ production. This potentially stimulates viruliferous *R. dorsalis* to probe more frequently and thus secrete more saliva into rice plants, thereby enhancing viral horizontal transmission.

We have discussed these on lines 344-356.

-Last sentence: it is bizarre to imagine “targeting CBP secretion” to combat viral transmission after showing that CBP actually blocks transmission. Do you mean increasing the CBP secretion to combat transmission? If yes, please explicitly write it.

Response: We have deleted the last sentence in discussion section.

Materials and Methods

-Line 481: Thirty second instar nymphs? What does it mean ? May be I am just ignorant on this point but over thirty larval stages seems a lot to me. Just checking.

-Line 492: same question

-Line 516 IDEM

Response: It means that there are 30 leafhoppers in each biological repetition. We have modified the relative descriptions in the full text.

Reviewer #2 (Remarks to the Author):

The authors report a dual action of virus infection on the saliva protein CBP. First, inhibition by an unidentified (viral?) mechanism lowers CBP levels in type-III salivary gland cells and promotes accumulation of virus via Pns11 filaments in salivary cavities. Second, lower CBP levels disturb normal 'stealthy' insect vector feeding behavior on rice plants and promote virus inoculation by provoking increased tissue punctures, caused by onset of plant defense reactions that interfere with feeding. The authors propose that by preventing calcium accumulation, CBP normally inhibits downstream plant defense responses against phloem feeding, in particular callose deposition and ROS formation, and that this inhibitory effect is elevated by the reduced CBP levels in infected insects. Indeed, lower CBP levels are found at feeding sites of viruliferous insects, and silencing CBP has the same effect.

This is a sound piece of work and the authors present ample evidence to back up their conclusions. The only 'brick' missing is how RGDV inhibits CBP, but the authors' intention is not to unveil this mechanism but to describe its effect on virus-vector-host interactions. The take home message 'viruses manipulate vector an insect saliva effector to induce insect feeding behavior conducive for virus transmission' is intriguing.

Response: Thank you for your positive evaluation of our work.

I have some concerns, though:

1. The authors claim that calcium accumulation is significantly higher at feeding sites from infected or CBP-silenced hoppers (lines 215ff). In my opinion, the images shown in Figure 5a alone do not allow to draw this conclusion. Quantification of the calcium signal taking into account fluorescence intensity/areas of feeding sites and using an adequate amount of feeding sites would be needed to back this claim. Further, as outlined below, the data do not allow to conclude whether you observe calcium signaling.

Response: We have quantitatively analyzed the fluorescence intensity of calcium accumulation in the feeding sites in different experimental groups, including 30 independent feeding sites for 1 h and 90 independent feeding sites for 3 h. In general, calcium signals were clearly observed in the cells around leafhopper feeding sites. The quantification analysis of fluorescence intensity was made using Image J

software. The new figures were shown in Figure 5c.

2. Whereas Figure 5c shows increased callose deposition in the vessels, I am not sure that the images show callose deposition on sieve plates, because they are practically invisible in transversal sections. For this, one needs to observe longitudinal sections, which allow to localize and count callose deposition on sieve plates. This applies also to Figure 5d. Please clarify whether you observed and quantified 'general' callose deposition in vessels or sieve plate-specific callose deposition. I guess both deposition forms would interfere with feeding behavior.

Response: We have added the new images of the longitudinal section and whole leaf section to clearly show the callose deposition using bright blue fluorescence. We calculated the callose deposition in the sieve plates of plants. The relative descriptions were shown on lines 285-289 and in Figure 5e.

3. The presentation of CBP is a bit short. It would be nice if you describe how you identified this protein and what is its relationship to other CBP from *R. d.* and other phloem feeders.

Response: Our preliminary transcriptome sequencing data analyses of *R. dorsalis* indicated that the expression of CBP in RGDV-infected salivary glands was significantly decreased relative to uninfected controls. The relative sentence was shown on lines 109-111.

CBP is a universal component in the saliva of these phloem feeders, which would facilitate continuous feeding by inhibiting sieve tube plugging in plant phloem. The green rice leafhoppers, aphids and planthoppers secreted salivary CBPs with the conserved EF-hand motifs in their saliva, though the CBPs from different insect species have low amino acid sequence identities. The relative sentences appeared on lines 360-365.

Other remarks:

Some confocal images are overexposed.

Response: We have adjusted and replaced the overexploded figures.

Line 60: It is not one stylet; the mouthparts of these insects are formed of 4 stylets that together form a needle-like structure comprising the food and saliva canal. So it is more adequate to refer to as 'stylets'.

Response: It has been changed as suggested in the full text.

Line 145: You mention here that in phloem infested with infected hoppers less CBP is detected but the western blot Figure 2j shows the contrary result. Please correct.

Response: We have corrected this in Figure 2j.

Line 203: You write here that more virus is detected in rice phloem when CBP-silenced infected hoppers are used to infest plants, but the legend to Figure 4e does not mention what the blot shows. Please correct.

Response: We have revised this in Figure 4e.

Line 218: You see here Ca accumulation; whether these represent Ca signals can not be concluded from the data shown.

Response: We have quantitatively analyzed the fluorescence intensity of calcium accumulation in the feeding sites in different experimental groups, including 30 independent feeding sites for 1 h and 90 independent feeding sites for 3 h. In general, calcium signals were clearly observed in the cells around leafhopper feeding sites. The quantification analysis of fluorescence intensity was made using Image J software. The new figures were shown in Figure 5c.

Line 248ff: The figure does not show frequencies (events/time) of waveforms, only their accumulated duration. However, Figure 5b shows that more feeding sites are established over time, which can be interpreted as a higher frequency of punctures. Please explain what exactly you mean by 'active ingestion from phloem'?

Response: Our EPG assay results showed that the inhibition of RdCBP expression potentially caused viruliferous vectors to require more probing attempts in the process of feeding and encounter more obstacles in the process of reaching the phloem. Thus, viruliferous vectors prolonged salivary secretion and active ingestion probing, further confirming the above result that viruliferous and dsRdCBP-treated leafhoppers established more number of feeding sites (Figure 5b). 'active ingestion from phloem' has changed as 'active ingestion from the phloem with the stylets' in the revised version. The relative sentences appeared on lines 254-260.

Line 320: Phloem sap is not really nutritionally rich; it is a rather unbalanced diet requiring adaptation of phloem feeders.

Response: We have revised this.

Lines 386-393: Please indicate briefly how you prepared total RNA and protein extracts from insects and/or plants, and the oligonucleotides used.

Response: Total RNAs and proteins were extracted from insect and/or plant using Trizol reagent (Invitrogen) and RIPA lysis buffer (ThermoFisher) according to the manufacturer's instructions. The relative sentences appeared on lines 392-396.

The oligonucleotides primers used in this paper were listed in Supplementary Table 1.

Line 401: Please indicate more in detail how you carried out immunofluorescence and acquired microscopic images or cite corresponding publications.

Response: The detailed experimental process for immunofluorescence and acquired microscopic images has been added on line 413-424.

Line 421: Reference (1) is not very useful to describe the methodology because it refers to yet other references.

Response: The new reference has been updated.

Line 451: Here you use a his and GST antibodies for protein detection, in the legends to the corresponding experiments you use GST, his, actin, CBP and Pns11 antibodies. Please verify.

Response: We have revised this on lines 786-787.

Line 469: Are these final concentrations?

Response: Yes, this is the final concentration after dilution.

Line 790: (e) the label of the blots is missing.

Response: The label of the blots has been added.

Line 545: What do you mean by 'freezing glue'?

Response: Freezing glue is an embedding medium for cryotomy, and specific product information has been added in lines 556-559.

Line 559: Who is the provider of the kit and is it fluorescence-based or colorimetric?

Response: Amplex® Red Hydrogen Peroxide/Peroxidase Assay Kit (Thermo Fisher Scientific, Molecular Probes™, catalog number: A22188) was used to detect hydrogen peroxide by fluorescence method. The specific product information has been added in line 571-573.

Line 885: d-f. In 3Sd, actin and virus images are switched.

Response: We have corrected this.

In some figures, 'levels' is misspelled, in Figure 7A 'mock' is misspelled.

Response: The spelling error of 'level' in Figures 1b, 2a, 4b and 4c have been corrected; The spelling error of 'mock' in Figure 7a has been corrected

Figure 1C: Please label with which antibody the blots were incubated.

Response: The label of the blots has been added.

Figure 2J: The text in the legend and in the blot diverge (actin in the blot, RGDV P8 in the legend). Why is the actin blot in blue? Please rectify.

Response: Figure 2J is Western blot assay of RdCBP in rice plants after viruliferous (V+) or nonviruliferous (V-) feeding (top). Rubisco large subunit was used as a loading control, as detected by staining with Coomassie Brilliant Blue (CBB) (bottom). We have modified the description in the Figure 2J legend.

Reviewer #3 (Remarks to the Author):

Insect vectors can transmit viral pathogens together with their saliva to plant phloem, but the roles of saliva components remain elusive. In this manuscript, Wu et al investigate the function of calcium-binding protein (CBP), a saliva protein of leafhopper, in the viral transmission by insect vectors. They found that the CBP competed with viral Pns11 filaments of rice gall dwarf virus (RGDV) to traverse the apical plasmalemma into saliva-stored cavities in salivary glands of leafhopper vectors. They further found that virus-caused reduction of CBP triggered substantial callose deposition and H₂O₂ production, which caused the stronger feeding barriers for viruliferous leafhopper vectors. More frequent probe and salivary secretion ultimately enhanced viral transmission.

I think this story is very interesting due to the relationship among salivary protein, pathogens, and insect behavior. These findings will benefit for understanding how salivary proteins can regulate the interaction between plant host, virus, and insect vector. of vector-borne viruses. However, I also have some questions that authors have to response.

Response: Thank you for your positive evaluation of our work.

1. Fig. 2q-s. Salivary glands were only immunolabeled with RdCBP-specific antibodies. However, authors did not show that the apical plasmalemma-associated filamentous structure was formed by RGDV Pns11. The immunoelectron microscopy of Pns11 is suggested.

Response: The immunoelectron microscopy of Pns11 has been added in Figure S5.

2. Fig. 4, How did authors determine the injection doses of dsRNA in the RNAi experiment? Did different dose of dsRNA have different effect on RGDV infection in salivary glands?

Response: We finally chose a suitable injection concentration of dsRNA after observing the effects of different concentrations of dsRNAs on the mortality of leafhoppers. In general, the concentration higher than the suitable injection concentration did not have different effects on RGDV infection in salivary glands.

3. Fig. 5, How many insects were recorded in the EPG experiment? Please show in the figure legend and Methods.

Response: Thirty insects in each experimental group were recorded in the EPG experiment. It has been added in Figure legend and methods.

4. How RGDV replicated and assembled in the cells salivary glands? Authors should discuss this.

Response: Relative descriptions have been added on lines 88-92.

Minor comments:

1. The specificity of the RdCBP antibody should be shown in Materials and Methods.

Response: We have detected the specificity of the antibody, and the related descriptions were shown on lines 119-122.

REVIEWERS' COMMENTS:

Reviewer #1 (Remarks to the Author):

The modifications introduced by the authors in this revised version are all OK. I have the feeling that the authors have made substantial efforts on the style of results presentation and discussion, on details of methods, and on additional data to improve the manuscript along the lines suggested by reviewers.

For my part, I confirm that the reponses of the authors to my comments are perfectly satisfactory

Reviewer #2 (Remarks to the Author):

The revised manuscript is significantly improved.
The following items, however, require slight adaptations of the text:

Lines 205ff: I am not sure that you can talk of calcium "signals" or that feeding "activated" cytosolic calcium. However, it is very clear that calcium concentrations increase significantly. So as a fact you have increased calcium levels and you can discuss that they might present calcium signals.

Lines 224ff: I am also not sure that you show that calcium increase regulates callose deposition. For this, other experiments are required. However, you results allow you to correlate increased calcium with increased callose. So please try to reformulate the sentence accordingly.

Reviewer #3 (Remarks to the Author):

The authors have revised the MS according to my comments.

Reviewer #2 (Remarks to the Author):

The revised manuscript is significantly improved.

The following items, however, require slight adaptations of the text:

Lines 205ff: I am not sure that you can talk of calcium "signals" or that feeding "activated" cytosolic calcium. However, it is very clear that calcium concentrations increase significantly. So as a fact you have increased calcium levels and you can discuss that they might present calcium signals.

Lines 224ff: I am also not sure that you show that calcium increase regulates callose deposition. For this, other experiments are required. However, your results allow you to correlate increased calcium with increased callose. So please try to reformulate the sentence accordingly.

Response: Thanks for the comments. In the revised version, we have carefully revised the related sentences to follow the suggestions. The related sentences appear on lines 207-228, and on lines 348-349.